# Activation of endogenous retroviruses during brain development causes an inflammatory response

Marie E Jönsson[1] [iD], Raquel Garza[1] [iD], Yogita Sharma[1], Rebecca Petri[1] [iD], Erik Södersten[2], Jenny G Johansson[1], Pia A Johansson[1] [iD], Diahann AM Atacho[1], Karolina Pircs[1] [iD], Sofia Madsen[1], David Yudovich[3], Ramprasad Ramakrishnan[4], Johan Holmberg[2], Jonas Larsson[3], Patric Jern[5] [iD] & Johan Jakobsson[1,*] [iD]

## Abstract

Endogenous retroviruses (ERVs) make up a large fraction of mammalian genomes and are thought to contribute to human disease, including brain disorders. In the brain, aberrant activation of ERVs is a potential trigger for an inflammatory response, but mechanistic insight into this phenomenon remains lacking. Using CRISPR/Cas9-based gene disruption of the epigenetic co-repressor protein Trim28, we found a dynamic H3K9me3-dependent regulation of ERVs in proliferating neural progenitor cells (NPCs), but not in adult neurons. *In vivo* deletion of *Trim28* in cortical NPCs during mouse brain development resulted in viable offspring expressing high levels of ERVs in excitatory neurons in the adult brain. Neuronal ERV expression was linked to activated microglia and the presence of ERV-derived proteins in aggregate-like structures. This study demonstrates that brain development is a critical period for the silencing of ERVs and provides causal *in vivo* evidence demonstrating that transcriptional activation of ERV in neurons results in an inflammatory response.

**Keywords** brain development; CRISPR; microglia; transposable elements; Trim28

**Subject Categories** Immunology; Neuroscience

The EMBO Journal (2021) 40: e106423

## Introduction

About one–tenth of the human and mouse genomes is made up of endogenous retroviruses (ERVs) (Jern & Coffin, 2008). This is a result of the cumulative infection of the germ line by retroviruses over millions of years. ERVs are dynamically silenced at the transcriptional level during early development via epigenetic modifications, including histone methylation and deacetylation as well as DNA methylation (Yoder *et al*, 1997; Rowe *et al*, 2010). Together, these repressive mechanisms suppress ERV expression in somatic tissues. However, it is becoming increasingly clear that ERVs are aberrantly activated in various human diseases, including a number of neurological disorders. For example, ERV expression has been found to be elevated in the cerebrospinal fluid and in post-mortem brain biopsies from patients with multiple sclerosis, amyotrophic lateral sclerosis, Alzheimer's disease, Parkinson's disease, and schizophrenia (Perron *et al*, 1997; Garson *et al*, 1998; Andrews *et al*, 2000; Karlsson *et al*, 2001; Steele *et al*, 2005; MacGowan *et al*, 2007; Perron *et al*, 2008; Douville *et al*, 2011; Li *et al*, 2015; Guo *et al*, 2018; Sun *et al*, 2018; Tam *et al*, 2019b).

Aberrant activation of ERVs in the brain has been proposed to be directly involved in the disease process through a number of different mechanisms, including the activation of an innate immune response, direct or indirect neurotoxicity or by modulating endogenous gene networks (Saleh *et al*, 2019; Tam *et al*, 2019a; Jonsson *et al*, 2020). However, causal studies of ERV activation in the brain are challenging, since this phenomenon is difficult to model in the laboratory. Most experimental studies rely on ectopic expression of ERV-derived transcripts, often using xeno-overexpression at non-physiological levels, making it hard to interpret the results (see e.g, (Antony *et al*, 2004; Li *et al*, 2015)). Still, while the role of ERVs in neurological disorders remains unclear (Tam *et al*, 2019a), ERV activation may constitute a new type of disease mechanism that could be exploited to develop much needed therapy for these disorders. Direct experimental evidence on the mechanisms underlying ERV repression and the consequences of ERV activation in the brain is therefore needed.

We recently found that Trim28, an epigenetic co-repressor protein, silences ERV expression in mouse and human neural progenitor cells (NPCs) (Fasching *et al*, 2015; Brattas *et al*, 2017).

1 Laboratory of Molecular Neurogenetics, Department of Experimental Medical Science, Wallenberg Neuroscience Center and Lund Stem Cell Center, Lund University, Lund, Sweden
2 Department of Cell and Molecular Biology, Karolinska Institutet, Stockholm, Sweden
3 Division of Molecular Medicine and Gene Therapy, Department of Laboratory Medicine and Lund Stem Cell Center, Lund University, Lund, Sweden
4 Division of Clinical Genetics, Lund University, Lund, Sweden
5 Science for Life Laboratory, Department for Medical Biochemistry and Microbiology, Uppsala University, Uppsala, Sweden
*Corresponding author. Tel: +46 46 2224225; Fax: +46 46 2220559; E-mail: johan.jakobsson@med.lu.se

Trim28 is recruited to genomic ERVs via Krüppel-associated box-zinc finger proteins (KRAB-ZFPs), a large family of sequence-specific transcription factors (Imbeault *et al*, 2017). Trim28 attracts a multiprotein complex that establishes transcriptional silencing and deposition of the repressive histone mark H3K9me3 (Sripathy *et al*, 2006). Trim28 is highly expressed in the brain and has been linked to behavioral phenotypes reminiscent of psychiatric disorders (Jakobsson *et al*, 2008; Whitelaw *et al*, 2010; Fasching *et al*, 2015).

In this study, we have investigated the consequences of *Trim28* deletion in the developing and adult mouse brain. We found that while Trim28 is needed for the repression of ERVs during brain development, it is redundant in the adult brain. Our results demonstrate the presence of an epigenetic switch during brain development, where the dynamic Trim28-mediated ERV repression is replaced by a different more stable mechanism. Interestingly, conditional deletion of *Trim28* during brain development resulted in ERV expression in adult neurons leading to an inflammatory response, including the presence of activated microglia and ERV-derived proteins in aggregate-like structures. In summary, our results provide direct experimental evidence *in vivo* for a link between aberrant ERV expression in the brain and an inflammatory response.

# Results

## CRISPR/Cas9-mediated deletion of *Trim28* in mouse NPCs

To evaluate the consequences of acute loss of *Trim28* in NPCs, we used CRISPR/Cas9 gene disruption. We generated NPC cultures from Rosa26-Cas9 knock-in transgenic mice (Fig 1A) (Platt *et al*, 2014), in which Cas9-GFP is constitutively expressed in all cells. These Cas9-NPCs were transduced with a lentiviral vector expressing gRNAs (LV.gRNAs) designed to target either exon 3, 4, or 13 of *Trim28* (g3, g4, g13) or to target *lacZ* (control). The vector also expressed a nuclear RFP reporter gene (H2B-RFP). Cas9-NPCs transduced with LV.gRNAs were expanded for 10 days, at which point RFP expressing cells were isolated by FACS (Fig 1A). To assess gene editing efficiency, we extracted genomic DNA from the RFP$^+$ Cas9-NPCs and performed DNA amplicon sequencing of the different gRNA target sites. We found that all three gRNAs (g3, g4 and g13) were highly effective, generating indels at a frequency of 98–99% at their respective target sequences (Fig 1B). The majority of these indels caused a frameshift in the Trim28 coding sequence that is predicative of loss-of-function alleles (Fig 1B) resulting in a near complete loss of Trim28 protein (Fig 1C). These results demonstrate that CRISPR/Cas9-mediated gene disruption is an efficient way to investigate the functional role of *Trim28*.

## Acute loss of *Trim28* in mouse NPCs results in upregulation of ERVs

We next queried if acute *Trim28* deletion in NPCs influences the expression of ERVs and other transposable elements (TEs). We performed strand-specific 2 × 150 bp RNA-seq on LV.gRNA-transduced Cas9-NPCs and investigated the change of expression in different ERV families using a TE-oriented read quantification software, TEtranscripts (Jin *et al*, 2015), while individual elements were analyzed using a unique mapping approach. Both of these analyses revealed an upregulation of ERVs upon the CRISPR-mediated *Trim28*-KO in mouse NPCs (Figs 1D and E, and EV1A). We found 13 upregulated ERV families, including IAPs and MMERVK10C. Both IAPs and MMERVK10C are recent additions to the mouse genome, and these ERV families include many full length, transposition-competent elements with the potential to produce long transcripts and ERV-derived peptides. We confirmed increased transcription of MMERVK10C elements using quantitative RT–PCR (qRT–PCR) (Fig EV1B). We also investigated the expression of other classes of TEs such as LINE-1s and SINEs but found no evident evidence of significant upregulation. Thus, acute deletion of *Trim28* in NPCs causes transcriptional upregulation of ERVs.

*Trim28* attracts a repression complex containing the histone methyltransferase SETDB1 that deposits H3K9me3 (Sripathy *et al*, 2006). CUT&RUN epigenomic analysis (Skene & Henikoff, 2017) of this histone modification in NPCs revealed that *Trim28*-controlled ERVs were covered by H3K9me3. For example, almost all full length members of the MMERVK10C family were covered by H3K9me3 (Fig 1F). Notably, only a handful of these individual elements were transcriptionally activated after *Trim28*-KO. This suggests that *Trim28* binds to many full length ERVs in NPCs but is only responsible for transcriptional silencing of a small subset of them. However, these elements are highly expressed upon *Trim28* deletion.

TEs have the potential to change the surrounding epigenetic landscape and consequently influence the expression of protein coding genes in their vicinity (Fasching *et al*, 2015; Brattas *et al*, 2017; Chuong *et al*, 2017). Accordingly, we found, for example, that genes located in the close vicinity of an upregulated MMERVK10C element also displayed a significant upregulation (Fig EV1C). In some instances, this was due to the activated ERV acting as an

---

**Figure 1. CRISPR/Cas9-based deletion of Trim28 in NPCs results in upregulation of ERVs.**

A   A schematic of the workflow for *Trim28*-KO in mouse NPC cultures. Scale bars: embryo 1 mm, NPCs 20 μm.

B   Estimation of gene editing at the *Trim28* loci using NGS-sequencing of amplicons. Black bars indicate % of frameshift indels. Columns show an average of two biological replicates per guide RNA and error bars show mean ± SD.

C   Western blot confirmed the loss of Trim28 expression upon *Trim28*-KO in mouse NPCs.

D   RNA-seq analysis of the expression of TE families using TEtranscripts

E   The significantly upregulated TE families with a fold change larger than 0.5 upon *Trim28*-KO in mouse NPCs. The dashed line indicates significance.

F   RNA-seq analysis of the *Trim28*-KO and control samples, visualizing full length MMERVK10C elements (left panels) and CUT&RUN analysis of H3K9me3 in mouse NPCs (right panel). The location of the full length MMERVK10Cs is indicated as a thick black line under each histogram.

G   Example of transcriptional readthrough outside a full length MMERVK10C into a nearby gene.

Source data are available online for this figure.

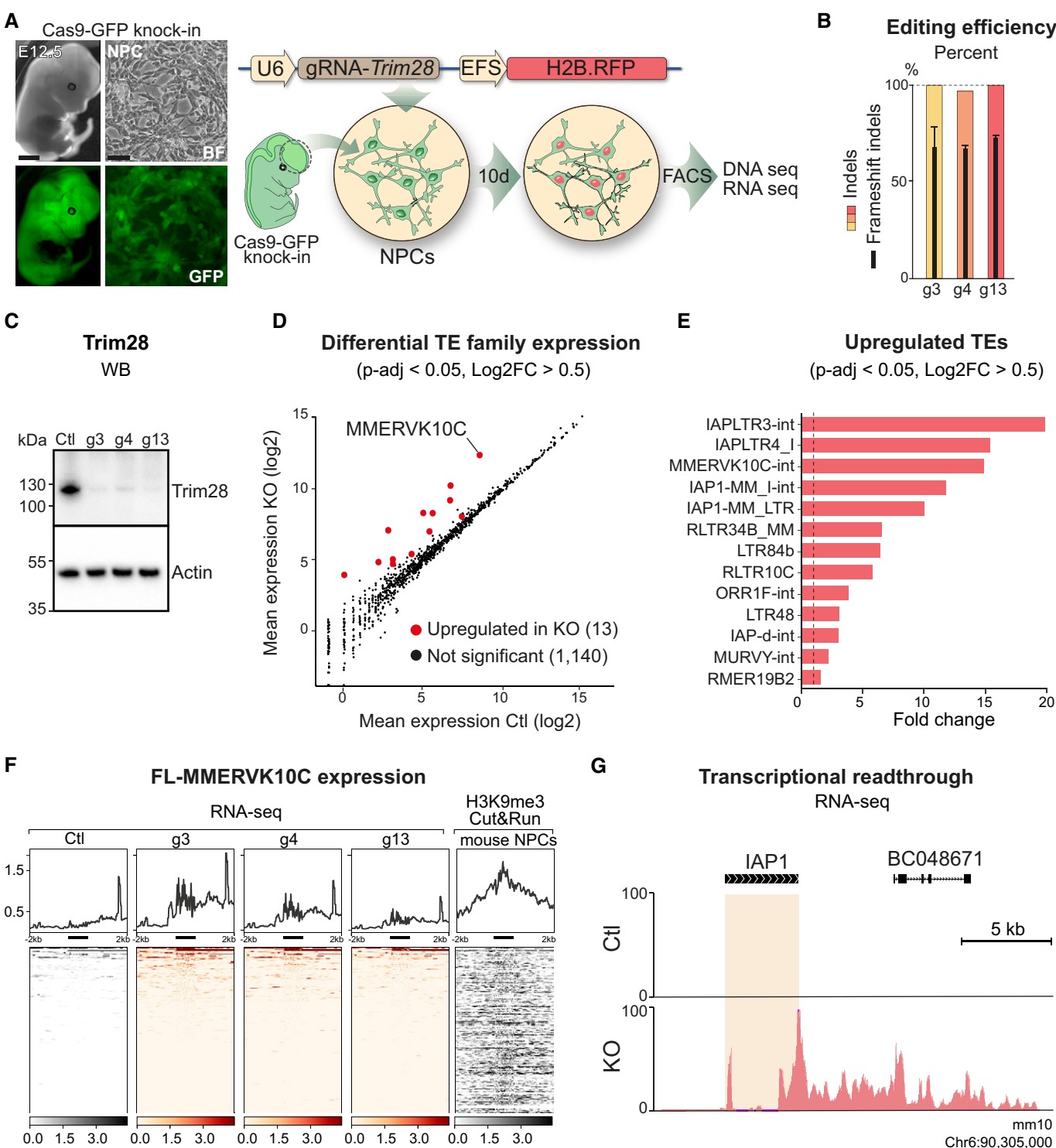

**Figure 1.**

alternative promoter (Fig 1G). Since Trim28 has additional functions in the cell (Quenneville *et al*, 2011) (Ziv *et al*, 2006) (Bunch *et al*, 2014), we also investigated the expression of all protein coding genes in *Trim28*-KO NPCs. This analysis revealed that acute loss of *Trim28* in NPCs had only a modest effect on protein coding genes (115 up- and 31 downregulated genes, respectively, Fig EV1D,

Table EV1). In addition, PCA analyses of differentially expressed protein coding genes and TEs revealed that the *Trim28*-KO cells separated from control cells based on TE expression rather than gene expression (Fig EV1E and F). Together, these results demonstrate that Trim28 silence the transcription of ERVs in NPCs but has a modest direct effect on protein coding genes.

## Deletion of *Trim28* in adult neurons *in vivo*

Alongside our previous data (Fasching *et al*, 2015; Brattas *et al*, 2017), these results confirm that *Trim28* is critical to repress ERVs in NPCs. However, it remains unclear how relevant these findings are to the situation *in vivo*. Therefore, we next investigated the consequences of deleting *Trim28* in mature neurons of adult mice. We designed adeno-associated viral vectors (AAV) vectors to drive expression of the *in vitro* verified gRNAs (AAV.gRNA) and used them separately or combined in the forebrain of Cas9-GFP knock-in mice. The AAV.gRNA vectors expressed H2B-RFP under the neuronal-specific synapsin promoter, allowing us to visualize and isolate transduced neurons (Fig 2A).

We injected AAV.gRNA into the forebrain of adult Cas9-GFP mice and sacrificed the animals after 8 weeks. The RFP expressing neuronal nuclei were isolated by FACS and amplicon sequenced to estimate the gene editing efficiency. All three gRNAs resulted in highly efficient gene editing (indel frequencies of 73–89%) where the majority of the indels were frameshift mutations (Fig 2B). Efficient deletion of *Trim28* was subsequently verified by immunohistochemistry (IHC) analysis, where quantification of Trim28 protein in RFP$^+$ cells showed loss of Trim28 expression in the majority of neurons (83–97%) in all of the groups (Fig 2C and D).

We next queried ERV expression in adult neurons lacking Trim28. We sequenced the RNA from the isolated RFP$^+$ nuclei and investigated the expression of ERV families as well as individual elements, using the same bioinformatical methods used for the NPCs. Remarkably, and in contrast to the NPC experiment, we observed no activation of ERVs upon *Trim28* deletion in adult neurons (Fig 2E). We also found no transcriptional activation of any other TE classes.

These results were particularly striking since Trim28 and many of its KRAB-ZFP adaptors are expressed in the brain, suggesting it is a significant organ for Trim28-mediated TE silencing (Imbeault *et al*, 2017). We therefore performed a series of additional control experiments to verify this finding. To ensure that the lack of ERV activation was not due to a bystander effect of nearby glial cells in which *Trim28* could be inactivated using our experimental setup, we developed an additional CRISPR-approach for cell type-specific deletion of *Trim28* in neurons using transgenic mice that conditionally express Cas9-GFP upon Cre expression (Stop-Cas9-GFP knock-in) (Platt *et al*, 2014) (Fig EV2A). We generated AAV vectors expressing the gRNAs and a Cre-inducible H2B-RFP reporter and an AAV vector expressing Cre under the control of the neuron-specific Synapsin1 (Syn) promoter. Upon transduction,

Cre expression and subsequent expression of RFP and Cas9-GFP resulted in highly efficient neuron-specific gene editing of *Trim28* (Fig EV2B–D). RNA-seq analysis for TE expression revealed no ERV activation upon *Trim28* removal (Fig EV2E), which were in line with our results from the ubiquitous Cas9-GFP knock-in mice. To further verify that the lack of ERV expression was not caused by potential Cas9-mediated side effects which may occur *in vivo*, we injected AAV vectors expressing Cre into the forebrain of adult floxed *Trim28* animals (Cammas *et al*, 2000) (Fig EV2F). Again, we obtained a highly efficient *Trim28* deletion in adult mouse neurons but did not observe ERV activation (Fig EV2G–I). Furthermore, ChIP-seq from adult mouse forebrain (Jiang *et al*, 2017) showed a lack of H3K9me3 accumulation on MMERVK10C sequences in adult neurons, in line with our observed lack of transcriptional activation upon *Trim28* deletion (Fig 2F). Taken together, these results demonstrate that Trim28 is not required to silence the transcription of ERVs in adult neurons.

## Deletion of *Trim28* in NPCs *in vivo*

Our results demonstrate that Trim28 is essential for transcriptional repression of ERVs in NPCs, but not in mature neurons. This suggests the existence of an epigenetic switch, where the dynamic and reversible Trim28/H3K9me3-mediated repression found in brain development is replaced by a different stable silencing mechanism in the adult brain. This is similar to what has been observed in early development where Trim28 participates in the establishment of DNA methylation to stable silence transposable elements (Wiznerowicz *et al*, 2007). To test this hypothesis, we deleted *Trim28* in dividing neural progenitors *in vivo*, which give rise to mature neurons in adulthood. We bred *Emx1*-Cre transgenic mice (Iwasato *et al*, 2000) with *Trim28*-flox mice (Cammas *et al*, 2000), resulting in *Trim28* deletion in cortical progenitors starting from embryonic day 10 (Figs 3A and EV3A). For better visualization of *Trim28*-excised cells by IHC, we included a *Cre*-inducible GFP reporter (gtRosa26-Stop-GFP) in the breeding scheme. With this setup, GFP expressing cells will correspond to cells in which Trim28 was deleted during development.

*Emx1*-Cre (+/−), *Trim28*-flox (+/+) mice were born at the expected ratio and survived into adulthood. Their overall brain morphology and size was not affected by the loss of *Trim28* during cortical development. IHC analysis of adult brains revealed that Trim28 protein was absent in virtually all pyramidal cortical neurons and that these cells also expressed GFP, demonstrating a highly efficient Cre-mediated excision of *Trim28* during development

---

**Figure 2. CRISPR/Cas9 deletion of Trim28 in adult neurons *in vivo*.**

A   A schematic of the workflow targeting *Trim28* in the mouse forebrain using AAV vectors expressing the gRNA and a nuclear RFP reporter. 8 weeks later, the injected animals were analyzed either by immunohistochemical analysis or nuclei isolation by FACS prior to DNA/RNA-sequencing.

B   Estimation of gene editing at the *Trim28* loci using NGS-sequencing of amplicons from DNA isolated from 50,000 RFP$^+$ nuclei per animal. One animal per group was analyzed. Black bars indicate % of the detected indels that disrupted the frameshift.

C, D   Gene editing of the *Trim28*-loci resulted in a robust loss of Trim28 protein, as evaluated by IHC where the expression of Trim28 in RFP$^+$ cells was quantified and is displayed as mean ± SEM. Approximately 600 RFP$^+$ cells per animal and group was evaluated. Scale bar 30 µm.

E   RNA-seq analysis of the expression of TE families using TEtranscripts.

F   RNA-seq analysis of the *Trim28*-KO and control samples, visualizing full length MMERVK10C elements (left panels) and ChIP-seq analysis of H3K9me3 in adult forebrain neurons (right panel). The location of the full length MMERVK10Cs is indicated as a thick black line under each histogram.

Source data are available online for this figure.

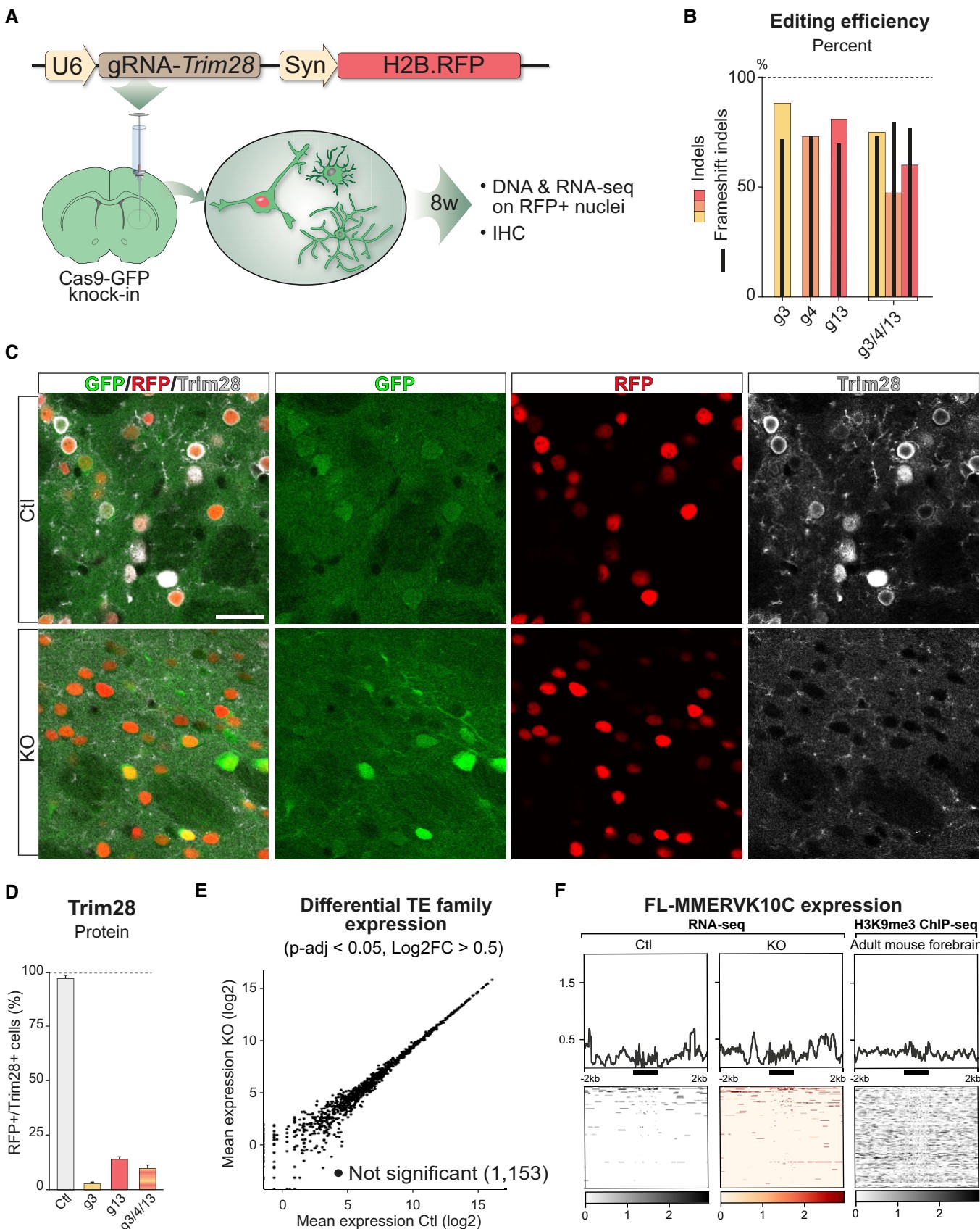

**Figure 2.**

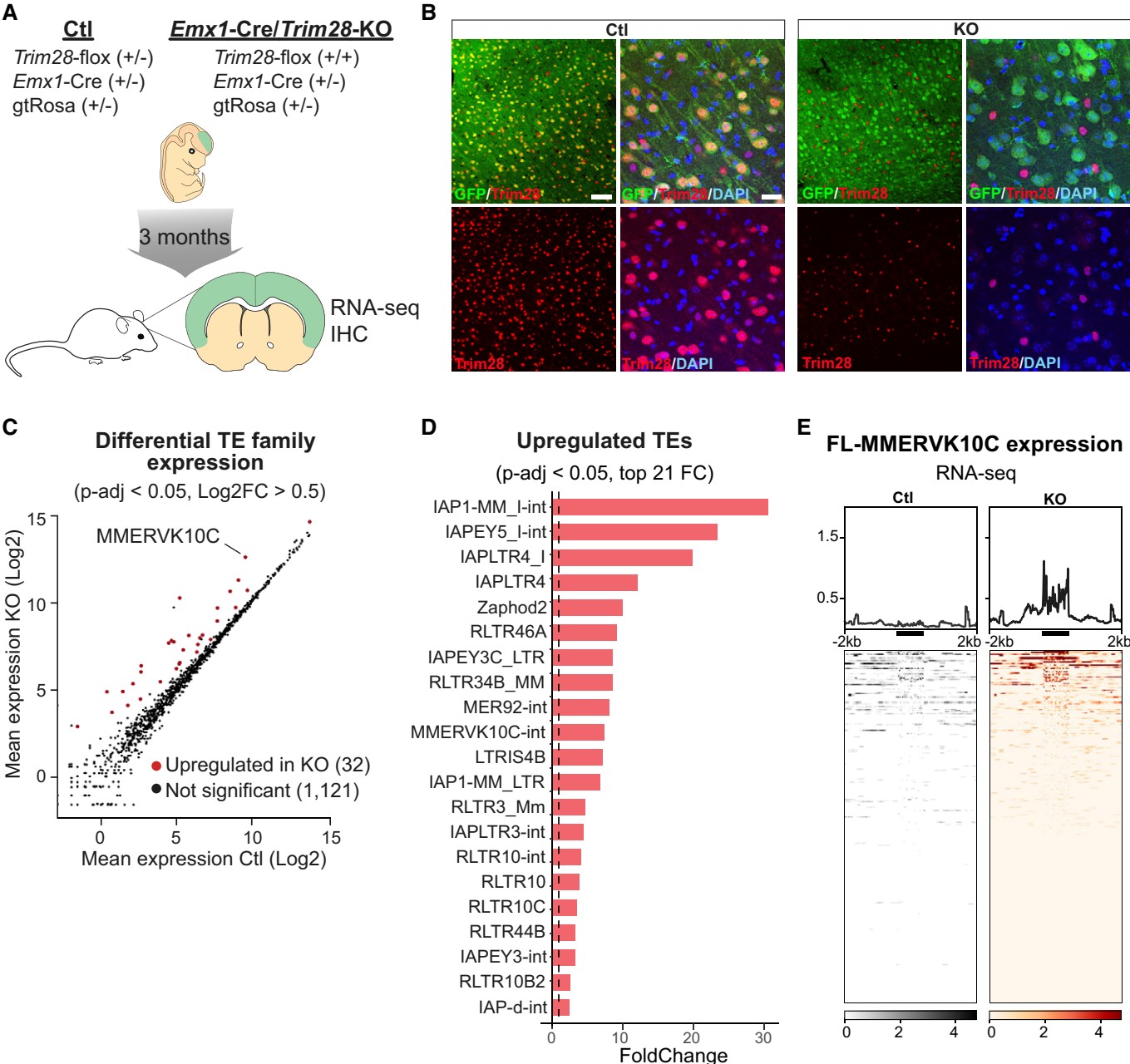

**Figure 3. Deletion of Trim28 during brain development results in aberrant TE expression in the adult brain.**

A  A schematic of the breeding scheme resulting in highly efficient conditional deletion of *Trim28*-KO during cortical development and analyzing the adult tissue 3 months later by IHC and RNA-seq.

B  IHC for Trim28 in the adult cortex revealed that the protein was lost in cells exposed to Cre-activity during brain development (GFP+ cells). Scale bars: low magnification 75 μm, high magnification 20 μm.

C  RNA-seq analysis of the expression of TE families using TEtranscripts

D  Significantly upregulated TE families upon the *Trim28*-KO, in which the families with the highest fold change are listed.

E  RNA-seq analysis of full length MMERVK10Cs in the adult tissue. The location of the full length MMERVK10Cs is indicated as a thick black line under each histogram.

(Figs 3B and EV3A). Cells that did not express GFP—for example, microglia and interneurons—expressed Trim28 as expected.

We performed RNA-seq on adult cortical tissue from these animals and control siblings to analyze ERV expression. In adult cortical tissue from *Emx1*-Cre (+/−), *Trim28*-flox (+/+), we observed a robust upregulation of several ERV families, many of which were also upregulated *in vitro* in the NPC experiment, including, e.g, MMERVK10C (Figs 3C and EV3B—individual TEs). Similar

to the NPC experiment, we found that only a few full length elements were activated, but that these were highly expressed (Fig 3D and E). Thus, the deletion of *Trim28* in NPCs *in vivo* during brain development results in the expression of ERVs in the adult brain. Notably, and in contrast to the *Trim28*-KO in NPCs *in vitro*, we did not detect an effect on nearby gene expression by activation of ERVs (Fig EV3C) nor observe any transcriptional readthrough from activated ERV elements into neighboring genes. This also included the very same elements that had this property *in vitro* in NPCs (Figs 3E and EV3D). These results demonstrate that deletion of *Trim28* during brain development *in vivo* results in high-level expression of ERVs in the adult brain.

### Downstream transcriptional consequences of ERV activation in the brain

Our finding that *Emx1*-Cre (+/−), *Trim28*-flox (+/+) mice survive—despite high levels of ERV expression in the brain—raises the question about the downstream consequences of ERV activation *in vivo*. We first compared the expression of protein coding genes in *Emx1*-Cre (+/−), *Trim28*-flox (+/+) mice to their control littermates. We included RNA-seq data from animals in which *Trim28* was deleted in adult neurons (AAV.Syn-Cre, *Trim28*-flox (+/+)) in this analysis since these samples provide an important control setting in which the loss of *Trim28* does not impact ERV expression but only other Trim28 targets (Fig EV2F–I). Analysis of the *Emx1*-Cre (+/−), *Trim28*-flox (+/+) mice revealed 164 significantly upregulated and 86 significantly downregulated genes, of which 26 of the upregulated and 13 of the downregulated were similarly changed in the AAV.Cre, *Trim28*-flox (+/+) mice (Fig EV3E–G, Table EV1). This demonstrates that the loss of Trim28 during brain development causes substantial downstream effects on gene expression. Gene ontology (GO) analysis on molecular and biological pathways of genes specifically altered in *Emx1*-Cre (+/−), *Trim28*-flox (+/+)—but not in AAV.Cre, *Trim28*-flox (+/+) mice—revealed significant changes in genes related to cell adhesion (Fig EV3H).

### Single-nuclei RNA-seq analysis of ERV-expressing brain tissue

These results indicate that the activation of ERVs in neurons results in downstream transcriptional effects that could have an impact on neuronal function. However, the brain is a complex tissue composed of several cell types located in close vicinity. To separate cell-intrinsic effects from cell-extrinsic effects, we performed single-nuclei RNA-seq analysis on forebrain cortical tissue dissected from *Emx1*-Cre (+/−), *Trim28*-flox (+/+) and control littermates. High-quality single-nuclei sequencing data were generated from a total of 14,296 cells, including 7,670 from *Emx1*-Cre (+/−), *Trim28*-flox (+/+) mice and 6,626 from control littermates (Fig 4A).

We first performed an unbiased clustering analysis to identify and quantify the different cell types present in the brain tissue. We detected seven different clusters (Figs 4B and EV4A–H), including excitatory and inhibitory neurons as well as several different glial populations, with excitatory neurons making up the largest cluster with more than half of the cells (Fig 4B). Overall, there was no major difference in cell number proportions between *Emx1*-Cre (+/−), *Trim28*-flox (+/+) and control animals (Fig 4C and D). For example, we found no reduction of excitatory neurons in the *Emx1*-

Cre (+/−), *Trim28*-flox (+/+) animals even though this cell population completely lacked *Trim28*, as demonstrated by IHC of Trim28 and GFP. We conclude that deletion of *Trim28* during brain development in neuronal progenitors does not result in significant cell death.

Next, we analyzed transcriptional differences between the two genotypes. Among the dysregulated genes in excitatory neurons, we found altered expression of several lncRNAs and protein coding genes (Table EV2), many of which are linked to neurological disorders. Notably, we observed reduced expression of *Hecw2*, a ubiquitin ligase linked to neurodevelopmental delay (Berko *et al*, 2017) and *Sgcz*, a transmembrane protein linked to mental retardation (Piovani *et al*, 2014) (Fig 4E). We also found transcriptional alterations in astrocytes and oligodendrocytes (Fig 4F and G), two additional cell types in which *Emx1* is expressed during brain development and therefore should lack Trim28 expression. Among the dysregulated genes in astrocytes, we detected upregulation of *Apoe*, the key risk variant gene for Alzheimer's disease, *Rora*, a nuclear hormone receptor linked to intellectual disability, epilepsy and autism (Guissart *et al*, 2018) and downregulation of *Auts2*, a transcriptional regulator linked to several neurodevelopmental disorders (Oksenberg & Ahituv, 2013). In oligodendrocytes, we observed transcriptional changes of *Fth1*, a ferritin gene linked to neurodegeneration (Muhoberac & Vidal, 2019), and *Ngr3*, a ligand to tyrosine kinase receptors that has been linked to schizophrenia (Kao *et al*, 2010). In contrast, interneurons, a neuronal subtype that maintains *Trim28* expression in *Emx1*-Cre (+/−), *Trim28*-flox (+/+) mice, displayed no evidence of altered gene expression in interneurons.

Interestingly, we noted that also microglial cells displayed transcriptome alterations (Fig 4H). For example, microglia showed a reduced expression of *Csmd1*, a complement regulatory gene linked to familial epilepsy (Naseer *et al*, 2016) and schizophrenia (Schizophrenia Psychiatric Genome-Wide Association Study & C, 2011), and of the protocadherin *Pcdh9*, which is a risk factor for major depressive disorder (Xiao *et al*, 2018). Similar to oligodendrocytes, microglia showed a downregulation of *Nrg3* (Fig 4H). Microglia are immune cells of endodermal origin that do not express Emx1 during development and therefore maintains *Trim28* expression in *Emx1*-Cre (+/−), *Trim28*-flox (+/+) animals. Taken together, these results demonstrate that the ERV activation in excitatory neurons, due to the loss of Trim28 during brain development, results in both cell autonomous and non-cell autonomous effects, specifically on microglia where some of the downstream dysregulated genes have been previously linked to psychiatric disorders.

### Cell type-specific analysis of ERV activation in the brain

The single-nuclei RNA-seq analysis indicated that microglia display transcriptional alterations despite maintaining Trim28 expression. Thus, microglia should be affected through cell-extrinsic mechanism mediated by derepressed ERVs from adjacent cells lacking Trim28. To verify this observation, we devised a strategy to analyze the expression of ERVs in the different cell populations in the *Emx1*-Cre (+/−), *Trim28*-flox (+/+) animals and control littermates. Since current pipelines available for high-throughput single cell RNA-seq analysis do not allow for estimation of TE expression, we developed an approach that initially uses the cell clusters established based on

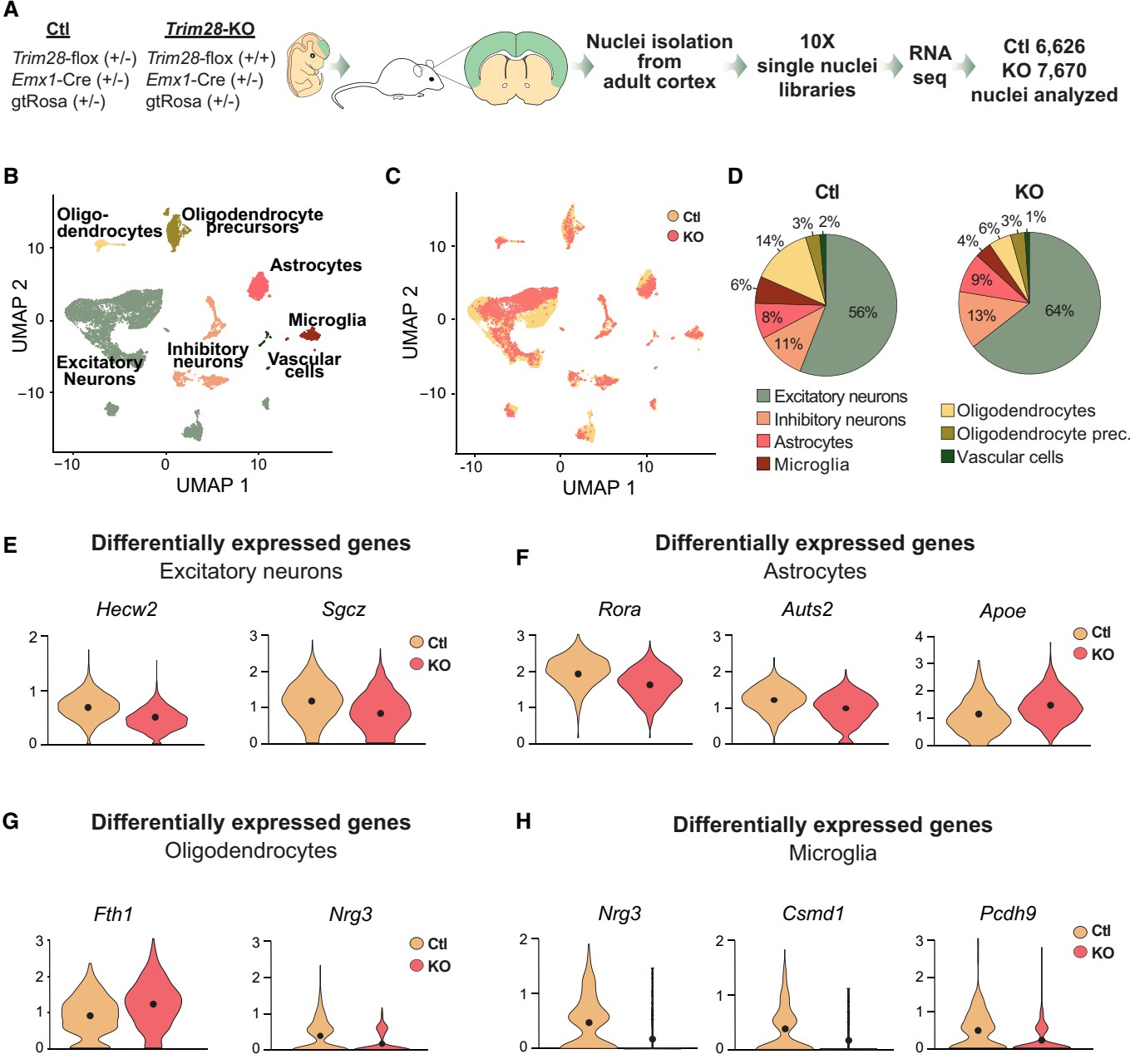

**Figure 4.  Single-nuclei RNA-seq of cortical tissue from *Emx1*-Cre/*Trim28*-KO animals and their littermates.**

A     A schematic of the workflow for the single-nuclei RNA-seq of cortical tissue from *Emx1*-Cre/*Trim28*-KO animals and their littermates.

B     UMAP showing the unbiased clustering analysis with seven different cell clusters.

C, D   UMAP and pie charts showing the distribution of *Trim28*-KO and control cells over the seven different clusters. There were no major differences in proportions of the different cell clusters between *Emx1*-Cre/*Trim28*-KO animals and controls.

E–H   A selection of significant cell-type-specific changes in gene expression between *Emx1*-Cre/*Trim28*-KO animals and controls as revealed by single-nuclei RNA-seq. The black dots represent the mean value (Wilcoxon rank sum test, (*P*-adj value < 0.01), *n* = 2. For a full list, see Table EV2.

the gene expression (Fig 4A and B). Then, by back-tracing the reads from cells forming each cluster we were able to analyze the expression of ERVs and other TEs using TEtranscripts in distinct cell populations (Fig 5A) (Jin *et al*, 2015), increasing the sensitivity of TE expression at single cell type-resolution.

When using this bioinformatical approach, we found that different ERV families (e.g, MMERVK10C) were expressed in excitatory neurons, astrocytes, oligodendrocytes, and oligodendrocytes progenitors, i.e, the cell types in which *Emx1* is expressed during brain development and thus lack Trim28 (Fig 5B). Notably, we

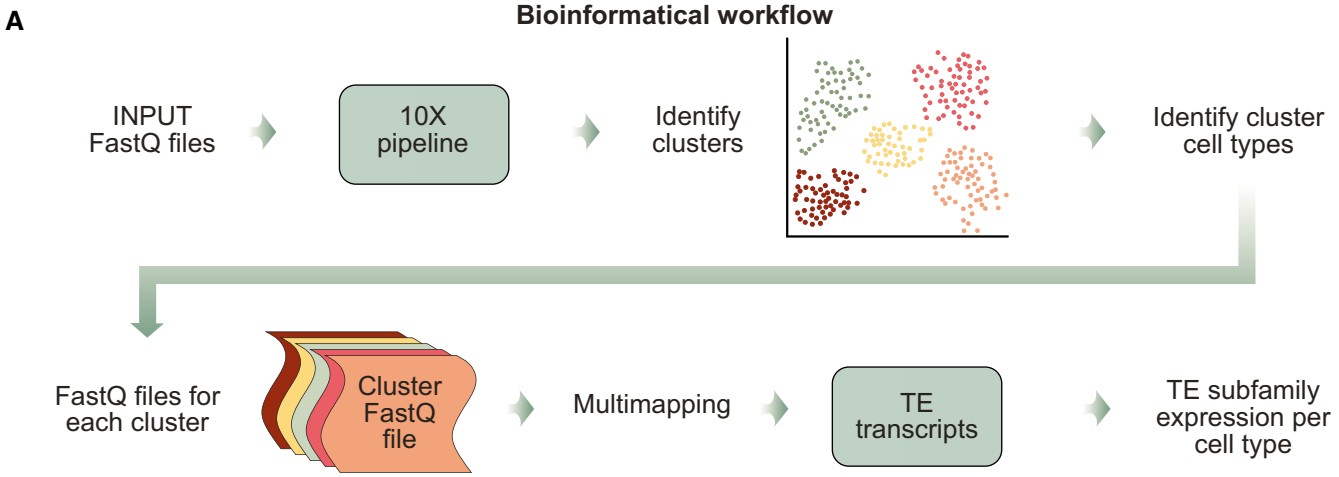

# Figure 5.

**Figure 5.  Cell type-specific analysis of ERV activation in *Emx1*-Cre/*Trim28*-KO animals.**

A   The workflow used to analyze ERV expression in single-nuclei RNA-seq data.
B   Mean plots show the changes of TE subfamily expression in each cell cluster upon the *Emx1*-Cre/*Trim28*-KO. A differential expression analysis performed with DESeq2 (described in material and methods) showed upregulated TEs in cell types in which *Trim28* was deleted (excitatory neurons, astrocytes, oligodendrocytes, and oligodendrocyte precursors), (indicated with red dots: *P*-adj < 0.01, log2FC > 3). The specific upregulated elements and their fold changes are listed in bar graphs under each mean plot. In cell types in which *Trim28* was not deleted, TE expression remained unaffected (inhibitory neurons and microglia).

found differences in ERV expression in distinct cell types, including activation of different families, demonstrating a cell type-specificity in the ERV-response to *Trim28* deletion. However, we found no upregulation of ERV expression in interneurons and microglia where Trim28 expression is maintained (Fig 5B). This analysis confirms that the transcriptional alterations observed in microglia are due to downstream cell-extrinsic effects.

### Expression of ERVs in the brain is linked to an inflammatory response

The microglia response to the *Trim28*-KO in neurons is intriguing as the aberrant expression of ERVs and other TEs have been linked to inflammatory responses (Hurst & Magiorkinis, 2015; Lim *et al*, 2015; Roulois *et al*, 2015; Thomas *et al*, 2017; Ishak *et al*, 2018; Saleh *et al*, 2019; Tam *et al*, 2019a). To study this further, we analyzed microglial cells by IHC for the pan-myeloid marker Iba1 (ionized calcium-binding adapter molecule 1, encoded by the Allograft inflammatory factor 1 (*Aif1*) gene). We found that the microglial cells in *Emx1*-Cre (+/−), *Trim28*-flox (+/+) mice displayed signs of activation, including higher expression of Iba1 (Fig 6A). Automated high-content screening microscopy analysis revealed that, although the density of Iba1 + cells was unaffected (Fig EV5A), the microglia present in the cortex of *Emx1*-Cre (+/−), *Trim28*-flox (+/+) mice displayed a morphology that is typical for an activation phenotype, including longer and thicker processes with increased numbers of branches (Fig 6B). Interestingly, we only found activated microglia in the cortex, where excitatory neurons lack Trim28, but not in the nearby forebrain structure striatum, where Trim28 is still expressed in all neurons (Fig 6C). The inflammatory response was therefore spatially restricted to the area with increased ERV expression. We also detected an increased expression of CD68, a lysosomal protein upregulated in activated microglia, in the cortex of *Emx1*-Cre (+/−), *Trim28*-flox (+/+) animals, a further indication of an ongoing inflammatory response (Fig 6D). In addition, we found no signs of gliosis or an inflammatory response in animals where *Trim28* was deleted in mature neurons, a setting where Trim28 is removed but no ERVs are activated (AAV.Syn-Cre/ *Trim28*-flox) (Fig EV5B). These results verify that the inflammatory response was not activated by the loss of *Trim28 per se* or by direct *Trim28*-targets in neurons, but more likely caused by the expression of ERVs.

### ERV-derived proteins are found in areas of microglia activation

A number of recent studies have demonstrated that the presence of ERV-derived nucleic acids can activate the innate immune system, such as double-stranded RNAs or single-stranded DNA (Hurst & Magiorkinis, 2015; Roulois *et al*, 2015; Ishak *et al*, 2018). According to this model, the host cells misinterpret the expression of ERVs as a

viral infection, and this triggers an autoimmune response through the activation of interferon signaling. Therefore, we searched for evidence of an interferon response and activation of a viral defense pathway in the transcriptome data (bulk RNA-seq and single-nuclei RNA-seq) from *Emx1*-Cre (+/−), *Trim28*-flox (+/+) mice. However, we found no evidence of activation of these pathways suggesting that the expression of genes linked to the innate immune response and viral response were not transcriptionally activated despite high levels of ERVs in the brain (Fig EV5C and D). Similarly, we found no evidence of this response in the *Trim28*-KO NPCs *in vitro*, in which ERVs were activated. Thus, deletion of *Trim28* and subsequent ERV activation in neural cells does not result in a detectable interferon response, suggesting that other cellular mechanisms are responsible for triggering the observed inflammatory response.

An alternative mechanism for ERV-mediated triggering of the inflammatory response is the expression of ERV-derived peptides and proteins, which could have neurotoxic properties (Li *et al*, 2015). To search for evidence of ERV-derived proteins, we performed WB and IHC analysis using an antibody detecting IAP-Gag protein in the brain of *Emx1*-Cre (+/−), *Trim28*-flox (+/+) mice. We found high levels of IAP-Gag protein expression in the brain of mice expressing elevated ERV transcripts, demonstrating their efficient translation into proteins (Fig 6E and F). The IAP-Gag labeling was restricted to cortical excitatory neurons lacking Trim28, as visualized by the co-expression of the Cre-dependent GFP reporter. Notably, the IAP-Gag labeling was not uniform, as some neurons expressed higher levels of IAP-Gag and some contained IAP-Gag in aggregate-like structures (Fig 6F), suggesting that the expression of ERVs in the brain results in the accumulation of ERV-derived proteins.

## Discussion

In this study, we define epigenetic mechanisms that control the expression of ERVs during brain development and investigate the consequences of their inactivation. Previous work has implicated ERVs and other TEs in several neurological disorders, such as MS, AD, ALS, PD, and schizophrenia, where an increased expression of TEs has been reported along with the speculation of their contribution to the disease process (Perron *et al*, 1997; Garson *et al*, 1998; Andrews *et al*, 2000; Karlsson *et al*, 2001; Steele *et al*, 2005; MacGowan *et al*, 2007; Perron *et al*, 2008; Douville *et al*, 2011; Li *et al*, 2015; Guo *et al*, 2018; Sun *et al*, 2018; Tam *et al*, 2019b; Jonsson *et al*, 2020). However, these clinical observations have been difficult to interpret since the results are correlative. Although there are studies that indicate a causality between upregulated TEs and neurodegeneration using both *Drosophila* and mouse model systems (Li *et al*, 2012; Li *et al*, 2013; Krug *et al*, 2017; Guo *et al*, 2018; Kremer *et al*, 2019; Sankowski *et al*, 2019; Dembny *et al*,

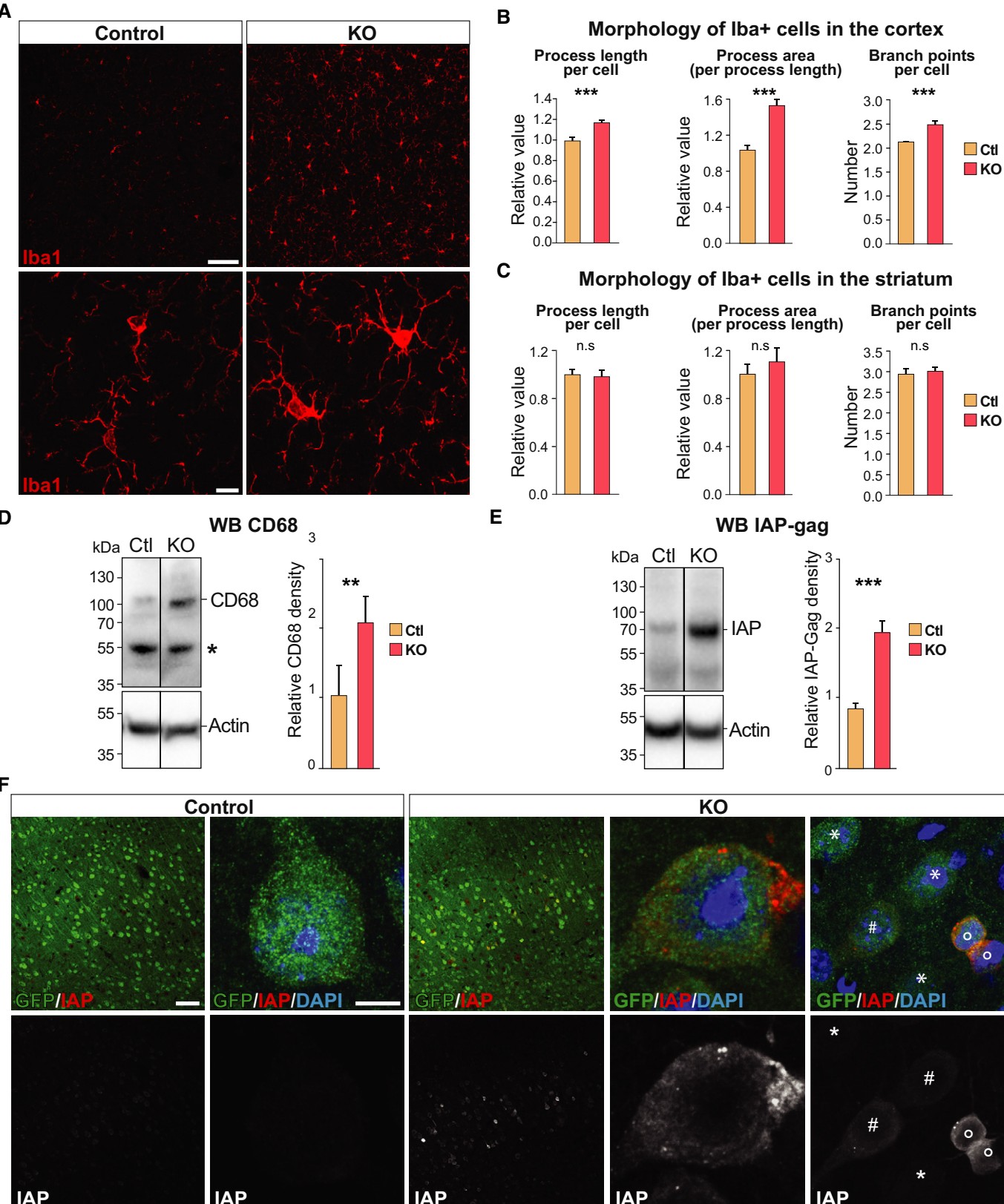

**Figure 6.**

◄

**Figure 6.  Activation of ERVs during brain development results in the presence of ERV proteins in adult brain tissue and signs of an inflammatory response.**

A   IHC analysis for the microglia marker Iba1 in *Emx1*-Cre/*Trim28*-KO animals and controls. Scale bars: Low magnification 75 µm, high magnification 10 µm.

B, C   The morphology of microglia in cortex and striatum (control region) of *Emx1*-Cre/*Trim28*-KO animals and controls were quantified by high-content screening of Iba1 immunoreactivity, revealing differences in process length, area, and number of branch points specifically in cortex, error bars mean ± SEM (unpaired *t*-test. *P* values: $1.8 \times 10^{-4}$, $9.6 \times 10^{-4}$, and $5.2 \times 10^{-6}$, respectively). A large number of photographs from three control and two KO animals were analyzed, see material and methods for details.

D   Western blot revealed an increased expression of CD68 in the *Emx1*-Cre/*Trim28*-KO animals (*n* = 5 per group), unpaired *t*-test, **P = 0.0036, error bars mean ± SEM. The star indicates an unspecific band.

E   An increased expression of the ERV-derived protein IAP-Gag was detected in the cortex of *Emx1*-Cre/*Trim28*-KO animals (*n* = 5 per group), unpaired *t*-test, ***P = 0.0008, error bars mean ± SEM.

F   Immunohistochemistry for IAP-Gag visualized the presence of ERV proteins in the cortex of *Emx1*-Cre/*Trim28*-KO animals. The distribution of IAP-Gag was heterogeneous among *Trim28*-KO neurons, where it was either accumulated in a low (#) or high (°) number of aggregate-like structures in the cytoplasm, or as weak homogenous staining throughout the cytoplasm (#) or not present at all (*). Scale bars: low magnification 75 µm, high magnification 5 µm.

Source data are available online for this figure.

2020), modeling of ERV activation in an experimental setting is challenging. Thus, the putative involvement of ERVs in brain disorders has remained controversial and direct experimental evidence on the consequences of ERV activation in the brain has been needed (Tam *et al*, 2019a; Jonsson *et al*, 2020). This study demonstrates that aberrant activation of ERVs during brain development *in vivo* triggers an inflammatory response linked to the presence of ERV-derived proteins present in aggregate-like structures in the adult brain. This increases our understanding of the consequences of ERV activation in the brain and provides new mechanistic insights opening up for further research into the role of ERVs and other TEs in various brain disorders.

Brain development is a critical period for the repression of ERVs and other TE, during which several chromatin-associated factors, such as DNMTs, HUSH, and Trim28/KRAB-ZFPs, work in parallel to mediate transcriptional silencing (Fasching *et al*, 2015; Brattas *et al*, 2017; Liu *et al*, 2018; Jonsson *et al*, 2019). Our data demonstrate that Trim28 dynamically controls the expression of ERVs through a mechanism linked to the repressive histone mark H3K9me3. Remarkably, we also found that Trim28 is required for the establishment of a different, stable silencing mechanism that is present in adult neurons—most likely DNA methylation (Wiznerowicz *et al*, 2007). Importantly, once established this system is independent of Trim28 and H3K9me3. This observation suggests that the ERV silencing machinery is particularly vulnerable during brain development and that perturbation of the system during this period could have life-long consequences. If Trim28-mediated silencing of ERVs is impaired during brain development, for example through mutations in KRAB-ZFPs or by environmental influence, ERVs may be derepressed in adult neurons, resulting in inflammatory response.

It is well established that many psychiatric disorders are a consequence of neurodevelopmental alterations (Bale *et al*, 2010; Horwitz *et al*, 2019). A combination of genetic components and environmental exposures are thought to contribute to the appearance and progress of these diseases, indicating that epigenetic alterations play a key role in the disease process (see e.g, (Khashan *et al*, 2008; Susser *et al*, 2008)). Many psychiatric disorders also have an immune component that is thought to play an important role in the disease process (Bauer & Teixeira, 2019). This immune response includes both peripheral activation and glial activation, in the brain. The underlying cause for this immune response remains largely unknown. However, the combination of epigenetic dysregulation and immune activation in psychiatric disorders, together with the observations made in the current study, suggest that ERVs are directly involved in the disease process. We have previously

demonstrated that the deletion of *Trim28* during postnatal forebrain development or heterozygous deletion of *Trim28* during early brain development results in complex behavioral changes including hyperactivity and an impaired stress-response (Jakobsson *et al*, 2008) (Fasching *et al*, 2015). In addition, heterozygous germ line deletion of *Trim28* in mice has been described to provoke an abnormal exploratory behavior (Whitelaw *et al*, 2010). These findings demonstrate that disruption of Trim28 levels in the mouse brain results in behavioral changes, similar to impairments found in psychiatric disorders. However, Trim28 is a multi-role protein which executes functions in the cell that is unrelated to the control of ERVs and we cannot formally exclude that such mechanisms contribute to the observed phenotypes (Ziv *et al*, 2006; Quenneville *et al*, 2011; Bunch *et al*, 2014). Still, aberrant ERV expression in settings which do not involve the loss of Trim28 has also indicated that ERV activation in the brain results in an inflammatory response as well behavioral impairments that are reminiscent to those observed in Trim28-mutant mice (Li *et al*, 2015; Kremer *et al*, 2019; Sankowski *et al*, 2019; Dembny *et al*, 2020; Jonsson *et al*, 2020).

In the current study, we found no evidence that activation of ERVs in NPCs, *in vitro* or *in vivo*, triggers an interferon response. This was unexpected, since a number of previous studies have demonstrated that activation of ERVs and other TEs through pharmacological inhibition of DNMTs, aging or by genetic alterations results in an interferon response through the recognition of double-stranded RNA or other TE-derived nucleic acids (Van Meter *et al*, 2014; Lim *et al*, 2015; Roulois *et al*, 2015; Thomas *et al*, 2017; Ahmad *et al*, 2018; De Cecco *et al*, 2019). On the other hand, there are mouse models, such as mice deficient for Toll-like receptors or antibodies (Young *et al*, 2012; Yu *et al*, 2012), that express high levels of ERVs without the appearance of an interferon response. Thus, the link between TE-activation and interferon response is likely context dependent, both in regard to the cell and tissue type as well as the identity of the TEs. Our results on ERV activation in the brain rather point to an alternative mechanism. One possibility is that ERV-derived peptides and proteins are implicated in the observed inflammatory response. We found numerous neurons displaying expression of ERV-derived proteins in the transgenic mice that expressed high levels of ERV-derived transcripts. These proteins were not distributed in a uniform pattern, but rather tended to form aggregate-like structures in a subset of neurons often located in close vicinity. This observation is interesting given the well-established link between protein aggregation and neurodegenerative disorders (Ross & Poirier, 2004). It is plausible that the presence of

ERV proteins directly or indirectly causes an inflammatory response, or they may serve as a trigger for further protein aggregation. Future in-depth studies are needed to understand this phenomenon.

In summary, these results demonstrate that Trim28 is required to silence ERVs during brain development and that the perturbation of this system results in an ERV-mediated inflammatory response in the adult brain. These results provide a new perspective to the potential cause and progression of neurodevelopmental and neurodegenerative disorders and further research into ERV-dysregulation in these types of disorders is therefore warranted.

# Materials and Methods

### Generation of Cas9-GFP mouse NPC cultures

All animal-related procedures were approved by and conducted in accordance with the committee for use of laboratory animals at Lund University.

The forebrain was dissected on embryonic day 13.5 from embryos obtained by breeding homozygote Cas9-GFP knock-in mice (Platt *et al*, 2014). The tissue was mechanically dissociated and plated in gelatin coated flasks and maintained as a monolayer culture (Conti *et al*, 2005) in NSA medium (Euromed, Euroclone) supplemented with N2 hormone mix, EGF (20 ng/ml; Gibco), bFGF (20 ng/ml; Gibco), 2 nM L-glutamine and 100 μg/ml Pen/Strep. Cells were then passaged 1:3–1:6 every 2–3 days using Accutase (Gibco).

### Targeting *Trim28 in vitro*

Guides were designed at crispr.mit.edu and are listed in the Appendix. Lentiviruses were produced according to Zufferey *et al*, (1997), and titers were $10^9$ TU/ml, which was determined using qRT–PCR. Cas9 mouse NPCs were transduced at a MOI 40 and allowed to expand for 10 days prior to FACS (FACSAria, BD Biosciences). Cells were detached and resuspended in basic culture media (media excluding growth factors) with propidium iodide (BD Biosciences) and strained (70 μm filters, BD Biosciences). RFP cells were FACS isolated at 4°C (reanalysis showed > 99% purity) and pelleted at 400 *g* for 5 min, snap frozen on dry ice and stored at −80°C until RNA/DNA were isolated. All groups were performed in biological triplicates.

### Targeting *Trim28 in vivo* using CRISPR/Cas9 in the adult brain

All animal-related procedures were approved by and conducted in accordance with the committee for use of laboratory animals at Lund University.

The production of AAV5 vectors has been described in detail elsewhere (Ulusoy *et al*, 2009), and titers were in the order of $10^{13}$ TU/ml, which was determined by qRT–PCR using TaqMan primers toward the ITR. Prior to injection, the vectors were diluted in PBS; the vectors containing the guide RNAs were diluted to 30% except upon co-injection of guides 3, 4, and 13 where the vectors were diluted to 10% each. Rosa26 Cas9 knock-in mice were anesthetized by isoflurane prior to the intra-striatal injections (coordinates from bregma: AP + 0.9 mm, ML + 1.8 mm, DV −2.7 mm) of 1 μl virus solution (0.1 μl / 15 s). The needle was kept in place for additional 2 min post-injection to avoid backflow. Animals were sacrificed

after 2 months and analyzed either by IHC or nuclei isolation (see details below) followed by DNA- or RNA-seq.

### Targeting *Trim28 in vivo* during neural development

Male *Emx1*-Cre (+/−); *Trim28*-flox (+/−); gtRosa (+/−) were bred with *Trim28*-flox (+/+) females to generate animals in which one (*Emx1*-Cre +/−; *Trim28*-flox +/−) or both (*Emx1*-Cre +/−; *Trim28*-flox +/+) *Trim28* alleles had been excised, used as control and *Trim28*-KO, respectively. Animals used for IHC were additionally heterozygote for gtRosa, in order to visualize the cells in which Cre had been expressed. Animals were genotyped from tail biopsies according to previous protocols (Cammas *et al*, 2000) and sacrificed at 3 months of age for either IHC or RNA-sequencing.

### Immunohistochemistry

Mice were given a lethal dose of phenobarbiturate and transcardially perfused with 4% paraformaldehyde (PFA, Sigma); the brains were post-fixed for 2 h and transferred to phosphate buffered saline (PBS) with 25% sucrose. Brains were coronally sectioned on a microtome (30 μm) and put in KPBS. IHC was performed as described in detail elsewhere (Sachdeva *et al*, 2010). Antibodies: Trim28 (Millipore, MAB3662, 1:500), Trim28 (Abcam, ab10484, 1:1,000), NeuN (Sigma-Aldrich, MAB377, 1:1,000), IAP-Gag (a kind gift from Bryan Cullen and described in (Dewannieux *et al*, 2004), 1:2,000), Iba1 (WAKO, no.019-19741, 1:1,000). All sections were counterstained with 4',6-diamidino-2-phenylindole (DAPI, Sigma-Aldrich, 1:1,000). Secondary antibodies from Jackson Laboratories were used at 1:400.

### Nuclei isolation

Animals were sacrificed by cervical dislocation and brains quickly removed. The desired regions were dissected and snap frozen on dry ice and stored at −80°C. The nuclei isolation was performed according to (Sodersten *et al*, 2018). In brief, the tissue was thawed and dissociated in ice-cold lysis buffer (0.32 M sucrose, 5 mM $CaCl_2$, 3 mM MgAc, 0.1 mM $Na_2$EDTA, 10 mM Tris–HCl, pH 8.0, 1 mM DTT) using a 1 ml tissue douncer (Wheaton). The homogenate was carefully layered on top of a sucrose cushion (1.8 M sucrose, 3 mM MgAc, 10 mM Tris–HCl, pH 8.0, and 1 mM DTT) before centrifugation at 30,000 ×*g* for 2 h and 15 min. Pelleted nuclei were softened for 10 min in 100 μl of nuclear storage buffer (15% sucrose, 10 mM Tris–HCl, pH 7.2, 70 mM KCl, and 2 mM $MgCl_2$) before resuspended in 300 μl of dilution buffer (10 mM Tris–HCl, pH 7.2, 70 mM KCl, and 2 mM $MgCl_2$) and run through a cell strainer (70 μm). Cells were run through the FACS (FACS Aria, BD Biosciences) at 4°C with low flowrate using a 100 μm nozzle (reanalysis showed > 99% purity). Sorted nuclei intended for either DNA or RNA-sequencing were pelleted at 1,300 ×*g* for 15 min and snap frozen, while nuclei intended for single-nuclei RNA-sequencing were directly loaded onto the 10× Genomics Single Cell 3′ Chip—see *Single-nuclei sequencing*.

### Analysis of CRISPR/Cas9-mediated *Trim28*-indels

Total genomic DNA was extracted from all *Trim28*-KO and control groups using DNeasy blood and tissue kit (Qiagen) and a 1.5 kb

fragment surrounding the different target sequences were amplified by PCR (see Table EV3 and EV4 for target and primer sequences, respectively) before subjected to NexteraXT fragmentation, according to manufacturer recommendations. Indexed tagmentation libraries were sequenced with 2 × 150 bp PE reads and analyzed using an in-house TIGERq pipeline to evaluate CRISPR/Cas9 editing efficiency.

### RNA-sequencing

Total RNA was isolated from frozen cell/nuclei pellets and brain tissue using the RNeasy Mini Kit (Qiagen) and used for RNA-seq (tissue pieces were run in the tissue lyser for 2 min, 30 Hz, prior to RNA isolation). Libraries were generated using Illumina TruSeq Stranded mRNA library prep kit (poly-A selection) and sequenced on a NextSeq500 (PE 2 × 150 bp).

The reads were mapped with STAR (2.6.0c) (Dobin *et al*, 2013), using gencode mouse annotation GRCm38.p6 vM19 as a guide. Reads were allowed to map to 100 loci with 200 anchors, as recommended by (Jin *et al*, 2015) to run TEtranscripts.

Read quantification was performed with TEtranscripts version 2.0.3 in multimode using gencode annotation GRCm38.p6 vM19 for gene annotation, as well as the curated GTF file of TEs provided by TEtranscripts authors (Jin *et al*, 2015). This file differs to RepeatMasker as it excludes simple repeats, rRNAs, scRNAs, snRNAs, srpRNAs, and tRNAs. The output matrix was then divided between TE subfamilies and genes to perform differential expression analysis (DEA) with DESeq2 (version 1.22.2) (Love *et al*, 2014) contrasting *Trim28*-KO against control samples. DESeq2 creates a general linear model assuming a negative binomial distribution using the condition of a sample and the normalized values of a gene. The resulting coefficients are tested between conditions using a Wald test. *P* values are then adjusted using Benjamini and Hochberg correction. For more information about the package methods, see (Love *et al*, 2014).

We report TE subfamilies as significantly different if their *P* adjusted value is below 0.05 and the absolute value of its log2 fold change is higher than 0.5.

To show the expression levels per condition, samples from the different guides targeting *Trim28* were pooled together and tested against the LacZ controls. The data were normalized using sizeFactors from the DESeq2 object (median ratio method described in (Anders & Huber, 2010) to account for any differences in sequencing depth.

In order to define differentially expressed elements and study their effects on gene expression, reads were uniquely mapped with STAR (2.6.0c). Full length mouse ERV predictions were done using the RetroTector software (Sperber *et al*, 2007), and read quantification of them was performed using featureCounts (Subread 1.6.3) (Liao *et al*, 2014). Differential expression analysis (DEA) was done with DESeq2. An intersection of the gencode annotation GRCm38.p6 vM19 with windows of 10, 20, and 50 kbp up and downstream of the upregulated elements was made with BEDtools intersect (Quinlan & Hall, 2010); same intersection was done for non-upregulated elements to compare their nearby gene dysregulation.

Bigwig files were normalized by RPKM using bamCoverage from deeptools and uploaded to USCS Genome Browser (release GCF_000001635.25_GRCm38.p5 (2017-08-04)).

Differential gene expression analysis was performed using DESeq2. Up- and downregulated genes (*P*-adj < 0.05, log2FC > 0.5) were used to test for GO terms overrepresentation using the web-based tool PANTHER (Mi *et al*, 2019). 30407594 We tested for overrepresentation of terms in their GO-Slim biological process dataset using Fisher's exact test with false discovery rates. Terms shown in main figures were those with more than four genes among the group of genes we were testing (up or downregulated), with an absolute log2 fold change value higher than 0.5 and a false discovery rate less than 0.05.

### Single-nuclei RNA-sequencing

Nuclei were isolated from the cortex of *Emx1*-Cre(+/−)/*Trim28*-flox (+/−, +/+) animals (Ctl *n* = 2, KO *n* = 2) as described above. 8,500 nuclei per sample were sorted via FACS and loaded onto 10× Genomics Single Cell 3′ Chip along with the Reverse Transcription Master Mix following the manufacturer's protocol for the Chromium Single Cell 3′ Library (10× Genomics, PN-120233) to generate single cell gel beads in emulsion. cDNA amplification was done as per the guidelines from 10× Genomics, and sequencing libraries were generated with unique sample indices (SI) for each sample. Libraries for samples were multiplexed and sequenced on a Novaseq using a 150-cycle kit.

The raw base calls were demultiplexed and converted to sample specific fastq files using cellranger mkfastq[1] that uses bcl2fastq program provided by Illumina. The default setting for bcl2fastq program was used, allowing 1 mismatch in the index, and raw quality of reads was checked using FastQC and multiQC tools. For each sample, fastq files were processed independently using cellranger count version 3.0 pipeline (default settings). This pipeline uses splice-aware program STAR[5] to map cDNA reads to the transcriptome (mm10). Since in nuclei samples it is expected to get a higher fraction of pre-mRNA, a pre-mRNA reference was generated using cellranger guidelines.

Mapped reads were characterized into exonic, intronic, and intergenic if at least 50% of the read intersects with an exon, intronic if it is non-exonic and it intersects with an intron and intergenic otherwise. Only exonic reads that uniquely mapped to transcriptome (and the same strand) were used for the downstream analysis.

Low-quality cells and genes were filtered out based on fraction of total number of genes detected in each cell (±3 nmads). From the remaining 16,671 nuclei, 6,472 came from control samples (Ctl) and 7,199 from knockout (KO).

For downstream analysis, samples were merged together using Seurat (version 3) R package (Dobin *et al*, 2013). Clusters have been defined with Seurat function FindClusters using resolution 0.1 and visualized with UMAP plots. Cell type annotation was performed using both known marker-based expression per cluster and a comparison of the expression profiles of a mouse brain Atlas (Zeisel *et al*, 2018). A marker gene set consisting of upregulated gene per cluster among the cells, combined with marker genes for all the 256 cell types in the atlas, was used in the comparison. The 256 atlas cell types were grouped into main clusters at Taxonomy rank 4 (39 groups), and mean expression per group was calculated using the marker gene set. These were compared to the mean expression in our clusters using Spearman correlation. Based on clusters annotation, clusters 0, 1, 2, 5, and 6, 7 were manually merged as excitatory

and inhibitory neurons, respectively. For each cell type, differential expression between knockout and control samples was carried out using Seurat function FindMarkers (Wilcox test, *P* adjusted < 0.01).

The expression of transposable elements was analyzed by extracting cell barcodes for all clusters using Seurat function Which-Cells, and the original.bam files obtained from the cellranger pipeline were used to subset aligned files for each cluster (subset-bam tool provided by 10×). Each.bam file was then converted back to clusters' fastq files using bamtofastq tool from 10×.

The resulting fastq files were mapped using default parameters in STAR using gencode mouse annotation GRCm38.p6 vM19 as a guide. The resulting bam files were used to quantify reads mapping to genes with featureCounts (forward strandness). The output matrix was then used to calculate sizeFactors with DESeq2 that would later be used to normalize TE counts.

The cluster fastq files were also mapped allowing for 100 loci and 200 anchors, as recommended by TEtranscripts authors. Read quantification was then performed with TEtranscripts in multimode (forward strand) using GRCm38.p6 vM19 for the gene annotation, and a curated GTF file of TEs given by TEtranscripts' authors. For further details, see the RNA-sequencing paragraph.

For the data presented in Fig 5B, the fold change bar plots were made from a DEA performed with DESeq2 of TE subfamilies of each cell type comparing control and knockout samples, for further details see the RNA-sequencing paragraph. The mean plots in the same figure were normalized using the sizeFactors resulting from the gene quantification with the default parameters' mapping.

## CUT&RUN

The CUT&RUN were performed according to (Skene & Henikoff, 2017). In brief, 200,000 mouse NPCs were washed, permeabilized, and attached to ConA-coated magnetic beads (Bang Laboratories) before incubated with the H3K9me3 (1:50, ab8898, Abcam) antibody at 4°C overnight. Cells were washed and incubated with pA-MNase fusion protein, and digestion was activated by adding CaCl$_2$ at 0°C. The digestion was stopped after 30 min and the target chromatin released from the insoluble nuclear chromatin before extracting the DNA. Experimental success was evaluated by capillary electrophoresis (Agilent) and the presence of nucleosome ladders for H3K9me3 but not for IgG controls.

The library preparation was performed using the Hyper prep kit (KAPA biosystems) and sequenced on NextSeq500 2 × 75 bp. Mapping of the reads to mm10 was performed with Bowtie2 2.3.4.2 (Langmead & Salzberg, 2012) using default settings for local alignment. Multi-mapper reads were filtered by SAMtools view version 1.4 (Li *et al*, 2009).

Using the ERVK prediction described in the section *RNA-sequencing*, we retrieved full length MMERVK10Cs. An ERVK was considered to be a full length MMERVK10C when an annotated MMERVK10C-int of mm10 RepeatMasker annotation (open-4.0.5—Repeat Library 20140131) would overlap more than 50% into the full length ERVK prediction. The intersection was performed with BEDtools intersect 2.26.0 (-f 0.5) (Quinlan & Hall, 2010). Heatmaps and profile plots were produced using deepTools' plotHeatmap (Ramirez *et al*, 2016) and sorted using maximum expression of the *Trim28*-KO samples or guide 3 for the *in vitro* and *in vivo* CRISPRs. Tracks for genome browser were

normalized using RPKM using deepTools' bamCoverage (version 2.4.3) (Ramirez *et al*, 2016).

The H3K9me3 ChIP-seq data from adult mouse cortex were retrieved from (Jiang *et al*, 2017), mapped, and analyzed in the same way as the in-house Cut & Run samples described above.

## qRT–PCR

Cortical brain pieces were disrupted in a tissue lyser (2 min, 30 Hz, 4°C) prior to RNA isolation using an RNeasy Mini Kit (Qiagen). cDNA was synthesized by the Maxima First-Strand cDNA Synthesis Kit (Invitrogen) and analyzed with SYBR Green I master (Roche) on a LightCycler 480 (Roche). Data are represented with the ΔΔCt method normalized to the housekeeping genes *Gapdh* and β-*actin*. Primers are listed in Table EV4.

## Western blot

Dissected cortical pieces from the *Emx1*-Cre (+/−); *Trim28*-flox (+/− and +/+) animals (Ctl *n* = 5, KO *n* = 5) were put in RIPA buffer (Sigma-Aldrich) containing Protease inhibitor cocktail (PIC, Complete, 1:25) and lysed at 4°C using a TissueLyser LT (Qiagen) on 50 Hz for 2 min, twice, and then kept on ice for 30 min before spun at 10,000 ×*g* for 10 min at 4°C. Supernatants were collected and transferred to a new tube and stored at −20°C. Each sample was mixed 1:1 (10 μl + 10 μl) with Laemmli buffer (Bio-Rad) and boiled at 95°C for 5 min before loaded onto a 4–12% SDS–PAGE gel and run at 200 V before electrotransferred to a membrane using Transblot-Turbo Transfer system (Bio-Rad). The membrane was then washed 2 × 15 min in TBS with 0.1% Tween20 (TBST), blocked for 1 h in TBST containing 5% non-fat dry milk, and then incubated at 4°C overnight with the primary antibody diluted in TBST with 5% non-fat dry milk (rabbit anti-Trim28, Abcam ab10484, 1:1,000; rabbit anti-CD68, 1:1,000, Abcam ab125212; rabbit anti-IAP-Gag, 1:10,000, a kind gift from Bryan Cullen and described in (Dewannieux *et al*, 2004)). The membrane was washed in TBST 2 × 15 min and incubated for 1 h in room temperature with HRP-conjugated anti-rabbit antibody (Sigma-Aldrich, NA9043, 1:2,500) diluted in TBST with 5% non-fat dry milk. The membrane was washed 2 × 15 min in TBST again and 1 × 15 min in TBS, before the protein expression was revealed by chemiluminescence using Immobilon Western (Millipore) and the signal detected using a ChemiDoc MP system (Bio-Rad). The membrane was stripped by treating it with methanol for 15 s followed 15 min in TBST before incubating it in stripping buffer (100 mM 2-mercaptoethanol, 2% (w/v) SDS, 62.4 mM Tris–HCL, pH 6.8) for 30 min 50°C. The membrane was washed in running water for 15 min, followed by 3 × 15 min in TBST before blocked for 1 h in TBST containing 5% non-fat dry milk. The membrane was then stained and visualized for β-actin (mouse anti-β-actin HRP, Sigma-Aldrich, A3854, 1:50,000) as described above.

## Morphological analysis

The morphology of Iba1[+] cells in the *Emx1*-Cre/*Trim28*-flox animals (Ctl *n* = 3, KO *n* = 2) was analyzed in 2D through an unbiased, automated process using the Cellomics Array Scan (Array Scan VTI, Thermo Fischer). The scanner took a high number of photographs

(using a 20× objective) throughout cortex (Ctl $n = 361$, KO $n = 215$) and striatum (Ctl $n = 104$, KO $n = 63$) and the program "Neuronal profiling" allowed analysis of process length, process area, and branchpoints per cell. 10 photographs of cortex from each animal were randomly selected, and Iba1[+] cells were manually counted in a blinded manner and presented as Iba1[+] cells per mm[2].

### Code availability

The pipeline, configuration files, and downstream analyses are available in the src folder at GitHub (https://github.com/ra7555ga-s/trim28_Jonsson2020.git). All downstream analysis and visualization were performed in R 3.5.1.

# Data availability

There are no restrictions in data availability. All file names are described in Table EV5, and the accession code for the RNA and DNA sequencing data presented in this study is GSE154196.

**Expanded View** for this article is available online.

## Acknowledgements

We would like to thank Molly Gale Hammell, Magdalena Götz, Sten Linnarsson, Chris Douse, Florence Cammas and Bryan Cullen for providing valuable reagents and input on the manuscript. We also thank, M. Persson Vejgården, U. Jarl, and A. Hammarberg for technical assistance. We are grateful to all members of the Jakobsson laboratory. The work was supported by grants from the Swedish Research Council (2018-02694, JJ & 2018-03017, PJ), the Swedish Brain Foundation (FO2019-0098, JJ), Cancerfonden (190326, JJ), Barncancerfonden (PR2017-0053, JJ), Formas (2018-01008, PJ) and the Swedish Government Initiative for Strategic Research Areas (MultiPark & StemTherapy).

## Author contributions

All authors took part in designing the study as well as interpreting the data. MEJ and JJ conceived and designed the study. MEJ, RP, JGJ, PAJ, DAMA, KP, SM, DY, RR performed experimental research and RG, PJ and YS performed bioinformatical analyses, JL ES, JH contributed resources. MEJ, RG, and JJ wrote the manuscript and all authors reviewed the final version.

## Conflict of interest

The authors declare that they have no conflict of interest.

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
