## [Review Process File · The EMBO Journal]

Activation of endogenous retroviruses during brain development causes an inflammatory response

Marie Jönsson, Raquel Garza, Yogita Sharma, Rebecca Petri, Erik Södersten, Jenny Johansson, Pia Johansson, Diahann Atacho, Karolina Pircs, Sofia Madsen, David Yudovich, Ramprasad Ramakrishnan, Johan Holmberg, Jonas Larsson, Patric Jern, and Johan Jakobsson

DOI: [10.15252/emboj.2020106423](https://doi.org/10.15252/emboj.2020106423)

Corresponding author(s): Johan Jakobsson (Johan.jakobsson@med.lu.se)

Review Timeline:

Submission Date:	4th Aug 20
Editorial Decision:	15th Sep 20
Revision Received:	29th Nov 20
Editorial Decision:	8th Jan 21
Revision Received:	22nd Jan 21
Accepted:	26th Jan 21

Editor: Karin Dumstrei

Transaction Report:

Dear Johan,

Thank you for submitting your manuscript to The EMBO Journal. Your study has now been seen by three referees and their comments are provided below.

As you can see from the comments, the referees find the analysis interesting. However, they also raise a number of good points that should be addressed. Should you be able to extend the analysis along the lines indicated then I would like to consider a revised version.

I am happy to discuss the raised points further and maybe it would be most helpful to do so via phone or video call.

Thank you for the opportunity to consider your work for publication. I look forward to your revision.

with best wishes

Karin

Karin Dumstrei, PhD
Senior Editor
The EMBO Journal

When assembling figures, please refer to our figure preparation guideline in order to ensure proper formatting and readability in print as well as on screen:
<http://bit.ly/EMBOPressFigurePreparationGuideline>

- a point-by-point response to the referees' comments, with a detailed description of the changes made (as a word file).

- a word file of the manuscript text.

- individual production quality figure files (one file per figure)

- a complete author checklist, which you can download from our author guidelines

(<https://www.embopress.org/page/journal/14602075/authorguide>).

- Expanded View files (replacing Supplementary Information)

Further information is available in our Guide For Authors:

The revision must be submitted online within 90 days; please click on the link below to submit the revision online before 14th Dec 2020.

Referee #1:

This manuscript reports on the role of Trim28 in repressing endogenous retroviruses during brain development, and also provides insight into the impacts of loss of such regulation. This is quite an interesting study that I think will generate interest in a broad audience. I also found that the findings are generally well documented and the experiments are well designed and presented. I am enthusiastic about this study, but have some critiques that I think the authors need to address before this is acceptable for publication.

First, some positives that I want to highlight. I very much enjoyed the fact that the authors of this study used methods to control the timing of Trim28 knock out, and also methods to control the cell types, and most impressively methods to examine both cell autonomous and non-autonomous expression responses. The findings with microglial activation are really interesting, and the fact that there are inflammatory responses without an interferon response also is quite important in my mind. A final highlight of this manuscript that I loved is that the authors found a creative way to get at TE expression within a single cell dataset. This is not easy, it in fact has been a real challenge because the signal typically is not up to the challenge in scSeq. So the approach of first finding cells that cluster by expression to define cell types followed by an examination of the reads from that population of cells is what I would call "sweet".

I do have several conceptual critiques.

First has to do with the potential pleiotropy of knocking out Trim28.

The authors make quite convincing arguments that the main impact is via ERV activation. I buy their arguments by and large. The effects on cellular genes are relatively minor, and it's quite cool that knock out in NPCs vs adult neurons is so different. This generally does support their claim. But I think the authors need to accept the possibility that Trim28 has some cellular targets and that the mechanism of its regulation of these is the same as for ERVs. In this case, when Trim28 is KO'd in NPCs, both ERV and cellular mRNA targets will be unregulated. But when it is KO'd in adult neurons, neither of these classes of targets will be up regulated. In this case, it remains possible that some of the effects seen from NPC KO could be contributed by some cellular targets.

This is not a deal killer to me. On balance, I think this is just a caveat that should be incorporated into the text in some way because it's very tough to rule out.

This same issue also impacts the discussion of the 'downstream effects' of ERV activation when Trim28 is KO'd in NPCs. For example the activation of genes related to cell adhesion might be a result of ERV expression, or they could be the result of cellular targets of Trim28 that also are developmentally regulated by Trim28 and then stably maintained by other mechanisms.

This same sort of caveat could apply to neuroinflammation, although in this case there are good reasons to believe that ERVs would have this impact.

Another critique that I think needs to be addressed is with regard to the IAP immunolabeling. The authors show very nice results with IAP gag, and it certainly has the look of aggregation. But this really isn't shown. I think the authors should either do biochemical experiments to show aggregates, or just remove that conclusion. They can speculate about aggregation, but shouldn't conclude it.

Third, I think that the fact that there is inflammation without interferon is very interesting. One thought I had, which might deserve comment, is that there may be differences in effects of ERV RNAs vs cDNAs. In the case of LINE elements, there is evidence that cDNAs accumulating in cytoplasm (Sedivy lab work for example) can activate cGAS/STING. On the other hand, accumulation of ERV RNAs would have different sensors. This may or may not activate interferon, so I wondered if that is worth comment or discussion?

Finally, in the discussion the authors correctly mention that there is evidence that ERVs and other retrotransposons may impact neurodegenerative phenotypes. And they are correct that showing causality in mammals has been very difficult, and this current dataset provides insight towards that goal. But worth mentioning that there is some good evidence for causality in flies both with TDP-43 and with Tau models.

I hope that the above comments are useful and will improve the manuscript.

Overall I am excited about the study.

Referee #2:

Reviewer comments for EMBOJ-2020-106423

Jönsson et al present an elegantly designed study that presents cell-type-specific effects of Trim28 loss. Using the different knockout systems the authors show that developmental deletion of Trim28 leads to sustained upregulation of certain ERVs. This effect is not found in adult neurons. Furthermore, the authors provide control experiments to corroborate their findings. On top of bulk-

RNA-Seq experiments, the authors conduct single-cell RNA Seq and even develop a custom algorithm for their analysis. Next, the authors go on to show that ERV activation leads to what they call "neuroinflammation".

All in all the study is timely and relevant. It is clearly written. The figures are designed intuitively. Experimentally, the authors do their due diligence by providing appropriate control experiments, including analysis of other retrotransposons and showing that Trim28-KO was shown upregulation specifically for specific ERVs. Unfortunately, the second aspect of the title regarding "neuroinflammation" is not well worked out. The authors mainly rely on unspecific morphological aspects of microglia. They would strengthen their point by collecting relevant functional readouts, such as microglia density, reduced expression of homeostatic proteins (Tmem119, P2ry12) or enhanced expression of activation markers (MHC II, CD68). Since the microglia phenotype is a major aspect of this study, the authors need to provide further experiments on that. They should also address the reviewer's misgivings about the choice of the term "neuroinflammation". The methods section needs additions and clarifications as specified below. Also, for a better appreciation of the study, the authors need to provide tables of differentially expressed genes.

Major Comments

1. The term "neuroinflammation" is too broadly applied. At histological levels the term "neuroinflammation" implies infiltration of peripheral immune cells into the brain parenchyma. Multiple sclerosis is considered the prototypic disease associated with neuroinflammation. In the current study, there is no evidence presented for infiltration of peripheral immune cells. To avoid confusion the authors should choose a term that more closely describes what is observed, such as "gliosis", or "microglia activation" or "inflammatory response".
2. The PCA-based comparisons of RNA-Seq datasets visualized in the figures S1d-e seem underpowered. If possible, please add 2-3 control samples in order to make a more robust conclusion from the data. This is important as the authors' claim regarding unchanged levels of protein-coding genes rests on it.
3. On page 15, the authors discuss the results of a differential gene expression analysis. Please provide the cutoffs for significance, multiple testing adjustment method and test used. Also, a table of the differentially expressed genes would be helpful. Furthermore, the biological processed term of cell adhesion is hardly "indicative of synaptic functions". To the reviewer's knowledge a large number of synapse function-related GO terms exist. To strengthen their biological interpretation, the authors should provide more thorough gene ontology term enrichment analysis (with IPA or similar software). Also, the reviewer might have missed it, but the authors do not seem to describe their enrichment analysis in the methods section. Please adjust.
4. On page 16, the authors claim that there were no differences in cell numbers between control and KO mice. However, this claim can only be made knowing the recombination efficiency of the Cre model. Could the authors please provide this data. Also, to appreciate the quality of the model and the data at hand, it would be helpful to see gene expression tSNE plots of Trim28.
5. Furthermore, the cell type classification of the scRNA-Seq data is based on the expression of marker genes. It would be helpful to see the cumulative expression of these genes projected on the tSNE map. Also, is there a reason the authors use tSNE? In the last 2 years UMAP has evolved as the standard visualization algorithm. Finally, the classification of microglia is based on genes that are also highly expressed in monocytes (Cd14, Cd68). Please provide evidence for the expression of microglia enriched genes (Tmem119, P2ry12, Hexb, Sall1) and Monocyte genes (Ly6c2, Fn1, Anxa2, Ccr2). Since your hypothesis is dependent on microglia, it is important to be more specific.

6. As for the method to obtain differentially expressed genes, was DeSeq2 used for the single cell data? Please specify in the methods section. It could make sense to use the differential gene expression functionality of Seurat. Here you can directly compare groups of cells that you define. For a fuller understanding of the paper please provide tables of all differentially expressed genes between control and KO mice and GO term analysis of these genes.
7. Is there a reason TrustER is not described in the methods section? Please adjust. Also at first mention on page 19 it's written "TrustTER".
8. The authors claim to have conducted "morphological analysis" of microglia. 2D analysis may not be a reliable readout. 3D reconstruction methods are available and should be used in such a setting. In case that the authors do not have access to such methods they should please clarify in the manuscript that they have used 2D morphological analysis and clearly state the shortcomings due to lack of 3-dimensional information.
9. On page 21, Iba1 is referred to as "inflammatory factor Iba1". This is misleading, please change to "the pan myeloid cell marker ionized calcium-binding adapter molecule 1 (Iba1) encoded by the Allograft inflammatory factor 1 (Aif1) gene". Please remove the statement that Iba1 is "specific" for microglia. It is just a pan-myeloid cell marker.
10. When claiming that microglia show "clear signs of activation" it is advisable to show downregulation of homeostatic microglia markers such as P2ry12 or Tmem119 and upregulation of microglial activation markers, such as MHC II and CD68. To support their claim and to back up the title of this study the authors need to provide these analyses.
11. Furthermore, microglia activation is associated with increased microglia densities. Please provide data if microglia counts per mm² change in the areas with ERV accumulation.
12. On page 24 the authors state that "This study provides direct in vivo evidence that aberrant activation of ERVs during brain development triggers neuroinflammation linked to ERV-derived protein aggregation in the adult brain." and they contrast it with correlation studies. However, at this point the authors only show that the presence of IAP-gag and microglia activation coincide, similar to, for example, Amyloid beta aggregates coinciding with Alzheimer's disease in patients. While it is tempting to make the direct connection, there are a number of alternative hypotheses. For example, microglia are activated due to aberrant neurophysiological activity of neurons expressing ERVs or that ERV expression triggers epileptogenic activity that microglia react to. The authors need to be realistic about the correlative nature of their findings and put it in context with relevant alternative hypotheses that they have not explored.
13. Several papers relevant to the topic of the current study have been published in the past 12 months (10.1073/pnas.1901283116, 10.1073/pnas.1822164116 and 10.1172/jci.insight.131093). The authors should please discuss these papers and how they relate to the present study.
14. On page 25, the authors should also put Trim28 in context with other molecular mechanisms with known roles in developmental ERV repression, such as Dnmt1, Dnmt3 (10.1016/j.ydbio.2009.07.017).
15. When discussing behavioral changes in Trim28-KO mice please put it in context with existing literature including the three studies mentioned under point 12.

16. On page 26, the authors note the lack of an interferon response during ERV activation. Similar observations were made in two landmark studies about ERV activation ([10.1016/j.immuni.2012.07.018](https://doi.org/10.1016/j.immuni.2012.07.018), [10.1038/nature11599](https://doi.org/10.1038/nature11599)). Please put your study in context with these.

Minor comments

It would have been helpful to provide a pdf document with numbered lines to facilitate the reviewing process.

Referee #3:

In this manuscript, the authors inactivate the Trim28/KAP1/Tif1beta co-repressor in mouse neuronal cells. In NPCs, in vitro, the inactivation results in increased transcription at a limited number of endogenous retroviruses (ERVs). A similar inactivation in adult neurons in vivo, interestingly, does not result in the reactivations of these ERVs, and the authors show that Trim28 needs to be inactivated during development to reach activity of the ERVs in the adult brain. This prompts the authors to suggest that Trim28, in association with Histone H3 lysine 9 methylation is required at the stem cell stage to allow for more stable repression of the ERVs later in life. This part of the study is followed up by an in-depth analysis at the single-cell level of ERV expression in the adult brain. Finally, the authors show that inactivation of Trim28 is associated with inflammation, possibly caused by strong expression and aggregation of proteins encoded by ERVs.

Overall, this study is thorough and contains several interesting observations. The main concern is that the interpretation of the data focuses exclusively on increased transcription of ERVs in the absence of Trim28, when in fact only a minute fraction of the ERVs get reactivated. Envisioning other possibilities as suggested below would greatly strengthen the paper. In addition, the last figures of the manuscript are somewhat descriptive, but this could probably be adjusted with minor additional experiments.

Figure 1 :

(1) A western blot showing the decreased Trim28 protein expression in the NPC KO cells is required.

(2) Panel 1C reports that 13 out of 1153 copies of MERVK10C are upregulated in the KO cells, while Sup Panel 1A shows that 122 TEs out of a total of approximate 3.5 million TE are upregulated in these cells. Likewise, in panel 1E, the heatmaps clearly show that it is only a very minor proportion of the FL-MMERK10C loci that are affected by the TRIM28 inactivation. First, RT-qPCR validation of some of the few activate loci would make the observation more robust. Next, given the very low proportion of activated ERVs, the authors may want to consider that the sequences they see activated have common traits other than being ERVs. For example, they may be sites of H3K9 methylation. This could easily be tested by doing peak calling on their Cut and Run data and then confront these peaks with the transcriptome data (as for FL-MMERK10C loci in 1E).

(3) Claiming that H3K9 methylation on FL-MMERK10C loci (as shown but the Cut & Run data) links these ERVs to Trim28 binding is a shortcut. This should be rephrased in the text.

(4) In the Sup. Figure 1, the inability of the PCA analysis in distinguishing WT from KO is not a very solid argument in favor of an absence of effect on gene expression. It is rather suggestive of poor reproducibility. This part should be rephrased.

(5) Examining the bigwigs shows that some of the usual targets of Trim28 do in fact get upregulated in the NPC Trim28 KO cells. The authors may want to look for example at the region

around Zfp991 (which incidentally is also extensively methylated on H3K9 according to the Cut & Run data). As this region gives strong Trim28 ChIP signal in other tissues, the effect of Trim28 inactivation on expression of genes in this region may be direct. Other similar regions may exist and could encode genes affecting the fate of the NPCs. This should be discussed.

Figure 3: Examining DNA methylation in the adult brain of the Emx1-Cre (+/-), Trim28-flox (+/+) mice would allow to confirm the hypothesis (Trim28 in stem cells defines DNA methylation later in development). This could be done genome-wide or just by checking some of activated TE by pyrosequencing. In addition, the authors claim that Sup. Figure 3 demonstrates that increased ERV transcription affects gene expression. This seems an overstatement, as the figure, strictly speaking, only shows that inactivation of Trim28 in NPCs affects gene expression later in life. This must be rephrased.

Figure 6: The presence of aggregates in adult tissues from adult Emx1-Cre (+/-), Trim28-flox (+/+) is an interesting observation. First, the authors need to verify that neither their primary nor their secondary antibodies stick to aggregates. Next, to gain mechanistic insight on how aggregates cause inflammation, the authors may want to check for activation of ER stress. The presence of ER stress is suggested by the increased expression of the ATF6 targets NR4A3 and EGR1 in the Emx RNA-seq data. To demonstrate this more directly, the authors could use their extracts from panel 6E with an anti-phosphoATF6 antibody to monitor maturation/cleavage of this protein.

Minor points:

- (1) Manual examination of the provided bigwigs showed increased transcriptional activity at several SINEs, which is in contrast with the description in the text. This could be due to poor management of the multi-mappers. Nevertheless, this should be checked.
- (2) RepeatMasker contains about 5 million TEs, Panel 1A examines only about 3.5 million TE. How were they selected?

Point-by-Point Jönsson et al.

Referee #1:

This manuscript reports on the role of Trim28 in repressing endogenous retroviruses during brain development, and also provides insight into the impacts of loss of such regulation. This is quite an interesting study that I think will generate interest in a broad audience. I also found that the findings are generally well documented and the experiments are well designed and presented. I am enthusiastic about this study, but have some critiques that I think the authors need to address before this is acceptable for publication.

First, some positives that I want to highlight. I very much enjoyed the fact that the authors of this study used methods to control the timing of Trim28 knock out, and also methods to control the cell types, and most impressively methods to examine both cell autonomous and non-autonomous expression responses. The findings with microglial activation are really interesting, and the fact that there are inflammatory responses without an interferon response also is quite important in my mind. A final highlight of this manuscript that I loved is that the authors found a creative way to get at TE expression within a single cell dataset. This is not easy, it in fact has been a real challenge because the signal typically is not up to the challenge in scSeq. So the approach of first finding cells that cluster by expression to define cell types followed by an examination of the reads from that population of cells is what I would call "sweet".

We would like to thank the reviewer for the overall positive comments on our manuscript. In the new version of the manuscript we have, in response to the reviewer's comments, added new data and modified the text. We honestly feel that this has resulted in a better manuscript that we now hope is ready for publication.

I do have several conceptual critiques.

First has to do with the potential pleiotropy of knocking out Trim28.

The authors make quite convincing arguments that the main impact is via ERV activation. I buy their arguments by and large. The effects on cellular genes are relatively minor, and its quite cool that knock out in NPCs vs adult neurons is so different. This generally does support their claim. But I think the authors need to accept the possibility that Trim28 has some cellular targets and that the mechanism of its regulation of these is the same as for ERVs. In this case, when Trim28 is KO'd in NPCs, both ERV and cellular mRNA targets will be unregulated. But when it is KO'd in adult neurons, neither of these classes of targets will be up regulated. In this case, it remains possible that some of the effects seen from NPC KO could be contributed by some cellular targets. This is not a deal killer to me. On balance, I think this is just a caveat that should be incorporated into the text in some way because its very tough to rule out.

This same issue also impacts the discussion of the 'downstream effects' of ERV activation when Trim28 is KO'd in NPCs. For example the activation of genes related to cell adhesion might be a result of ERV expression, or they could be the result of cellular targets of Trim28 that also are developmentally regulated by Trim28 and then stably maintained by other mechanisms.

This same sort of caveat could apply to neuroinflammation, although in this case there are good reasons to believe that ERVs would have this impact.

We appreciate that the reviewer raised this point. In the new version of the manuscript we have added a comment along these lines to the discussion (p. 18).

Another critique that I think needs to be addressed is with regard to the IAP immunolabeling. The authors show very nice results with IAP gag, and it certainly has the look of aggregation. But this really isn't shown. I think the authors should either do biochemical experiments to show aggregates, or just remove that conclusion. They can speculate about aggregation, but shouldn't conclude it.

We agree with the reviewer that the use of "aggregate" may have been misleading in the previous version of the manuscript. We have now changed the use of this word accordingly. In some instances, we have removed the word completely and in other parts of the text we refer to it as "aggregate-like structures".

Third, I think that the fact that there is inflammation without interferon is very interesting. One thought I had, which might deserve comment, is that there may be differences in effects of ERV RNAs vs cDNAs. In the case of LINE elements, there is evidence that cDNAs accumulating in cytoplasm (Sedivy lab work for example) can activate cGasSTING. on the other hand, accumulation of ERV RNAs would have different sensors. This may or may not activate interferon, so I wondered if that is worth comment or discussion?

We agree with the reviewer that this is a complex issue and that the literature is a bit inconsistent on this topic. We have added a new paragraph on this topic in the discussion in the new version of the manuscript (p. 18-19).

Finally, in the discussion the authors correctly mention that there is evidence that ERVs and other retrotransposons may impact neurodegenerative phenotypes. And they are correct that showing causality in mammals has been very difficult, and this current dataset provides insight towards that goal. But worth mention that there is some good evidence for causality in flies both with TDP-43 and with Tau models.

We thank the reviewer for pointing this out and we have added the causality between TEs and neurodegeneration shown in flies to the discussion on p. 16 and included suitable references.

I hope that the above comments are useful and will improve the manuscript. Overall I am excited about the study.

Referee #2:

Reviewer comments for EMBOJ-2020-106423

Jönsson et al present an elegantly designed study that presents cell-type-specific effects of Trim28 loss. Using the different knockout systems the authors show that developmental deletion of Trim28 leads to sustained upregulation of certain ERVs. This effect is not found in adult neurons. Furthermore, the authors provide control experiments to corroborate their findings. On top of bulk-RNA-Seq experiments, the authors conduct single-cell RNA Seq and even develop a custom algorithm for their analysis. Next, the authors go on to show that ERV activation leads to what they call "neuroinflammation".

All in all the study is timely and relevant. It is clearly written. The figures are designed intuitively. Experimentally, the authors do their due diligence by providing appropriate control experiments, including analysis of other retrotransposons and showing that Trim28-KO was shown upregulation specifically for specific ERVs.

Unfortunately, the second aspect of the title regarding "neuroinflammation" is not well worked out. The authors mainly rely on unspecific morphological aspects of microglia. They would strengthen their point by collecting relevant functional readouts, such as microglia density, reduced expression of homeostatic proteins (Tmem119, P2ry12) or enhanced expression of activation markers (MHC II, CD68). Since the microglia phenotype is a major aspect of this study, the authors need to provide further experiments on that. They should also address the reviewer's misgivings about the choice of the term "neuroinflammation". The methods section needs additions and clarifications as specified below. Also, for a better appreciation of the study, the authors need to provide tables of differentially expressed genes.

We are grateful for the overall positive comments on our manuscript. In the new version of the manuscript we have, as suggested by the reviewer, expanded the analysis of microglia and also modified the text accordingly. We hope that with these additions, our manuscript is now ready for publication.

Major Comments

1. The term "neuroinflammation" is too broadly applied. At histological levels the term "neuroinflammation" implies infiltration of peripheral immune cells into the brain parenchyma. Multiple sclerosis is considered the prototypic disease associated with neuroinflammation. In the current study, there is no evidence presented for infiltration of peripheral immune cells. To avoid confusion the authors should choose a term that more closely describes what is observed, such as "gliosis", or "microglia activation" or "inflammatory response".

We agree with the reviewer and have adjusted the terminology accordingly in the title and throughout the manuscript.

2. The PCA-based comparisons of RNA-Seq datasets visualized in the figures S1d-e seem underpowered. If possible, please add 2-3 control samples in order to make a more robust conclusion from the data. This is important as the authors' claim regarding unchanged levels of protein-coding genes rests on it.

We thank the reviewer for highlighting this. We have re-phrased the text on page 7 to clarify our conclusion from these two PCA-plots: "PCA analyses of differentially expressed protein coding genes and TEs revealed that the Trim28-KO cells separated from control cells based on TE expression rather than gene expression (Fig S1e-f). Together these results demonstrate that Trim28 robustly represses the transcription of ERVs in NPCs but has a marginal direct effect on protein coding genes."

We have also added a clarifying indication in SFig 1e-f:

3. On page 15, the authors discuss the results of a differential gene expression analysis. Please provide the cutoffs for significance, multiple testing adjustment method and test used. Also, a table of the differentially expressed genes would be helpful.

Differential gene expression analysis was performed using DESeq2. This tool creates a general linear model, assuming a negative binomial distribution, using the normalized counts of a gene and the condition of the sample. The resulting statistics are given by a Wald test. P values are adjusted for multiple testing using Benjamini and Hochberg correction. This description can now be found on p. 24 in the manuscript.

The cutoffs for significance between the conditions are indicated in Fig S3: if the p-adjusted value < 0.05 and the absolute value of the log₂ fold change > 0.5.

A table providing all significantly different gene expressions are now provided as Supplementary table 1&2 (bulk RNA-seq and single cell RNA-seq, respectively).

Furthermore, the biological processed term of cell adhesion is hardly "indicative of synaptic functions". To the reviewer's knowledge a large number of synapse function-related GO terms exist. To strengthen their biological interpretation, the authors should provide more thorough gene ontology term enrichment analysis (with IPA or similar software).

We agree with the reviewer that this statement was a bit misleading and has removed "indicative of synaptic functions". We have also attempted other types of enrichment analysis – but these gives no further insight into the phenotype.

Also, the reviewer might have missed it, but the authors do not seem to describe their enrichment analysis in the methods section. Please adjust.

We thank the reviewer for noticing that the enrichment analysis was missing from the methods section. It has now been added on p. 24-25. In short, we performed statistical overrepresentation with Fisher exact test in Panther using their GO-Slim biological process dataset. We plotted the terms which had more than four genes among our upregulated genes, had a log₂ fold change higher than 0.5 and less than 0.05 of FDR.

4. On page 16, the authors claim that there were no differences in cell numbers between control and KO mice. However, this claim can only be made knowing the recombination efficiency of the Cre model. Could the authors please provide this data. Also, to appreciate the quality of the model and the data at hand, it would be helpful to see gene expression tSNE plots of Trim28.

We have monitored the excision in the Emx1-cre model using two approaches:

- We have included *gtRosa28(+/-)* in the breeding scheme
- We have performed IHC for *Trim28*

Both these approaches indicate that excision is almost complete in cortical excitatory neurons. This data can be found in Fig 3b. There is also abundant excision in astrocytes and oligodendrocytes. There is no excision in microglia and interneurons. Based on these facts we are confident when we claim that viability of excitatory neurons is not affected by the deletion of *Trim28*.

Unfortunately, it is not valuable to plot *Trim28* expression using the single-nuclei RNA-seq data. In the *Trim28*-floxed animals, exons 4-14 are excised upon the presence of Cre (Cammass et al Development 2000 127: p. 2955-2963). Thus, the 3'-end of *Trim28*-transcripts can still be detected in 10x single cell RNA-seq data due to the 3' preference of this technology. However, the loss of the majority of the *Trim28* transcript is very clearly detected in the bulk RNA-seq (see Fig S2g).

5. Furthermore, the cell type classification of the scRNA-Seq data is based on the expression of marker genes. It would be helpful to see the cumulative expression of these genes projected on the tSNE map. Also, is there a reason the authors use tSNE? In the last 2 years UMAP has evolved as the standard visualization algorithm.

We thank the reviewer for the comments. As suggested, we have changed the projections of the expression of cell type specific genes to UMAP (Fig 4 and S4).

Finally, the classification of microglia is based on genes that are also highly expressed in monocytes (*Cd14*, *Cd68*). Please provide evidence for the expression of microglia enriched genes (*Tmem119*, *P2ry12*, *Hexb*, *Sall1*) and Monocyte genes (*Ly6c2*, *Fn1*, *Anxa2*, *Ccr2*). Since your hypothesis is dependent on microglia, it is important to be more specific.

We thank the reviewer for suggestions to clarify the specificity of the microglia cluster and have changed Fig S4 to projections of gene expression on UMAP instead of bar plots.

We have looked into the additional microglia markers suggested by the reviewer and have added the UMAP projections of *Tmem119* and *P2ry12* to Fig S4. The projection of *Hexb1* is shown below (but left outside the manuscript). We did not detect *Sall1* in our data set.

We detected two of the suggested monocyte genes in our data set (*Fn1* and *Anxa2*) and have included their UMAP projections in Fig S4 in order to visualize the cluster of vascular cells.

6. As for the method to obtain differentially expressed genes, was DeSeq2 used for the single cell data? Please specify in the methods section. It could make sense to use the differential gene expression functionality of Seurat. Here you can directly compare groups of cells that you define.

We appreciate reviewer's comment on this. Differential gene expression analysis in the single cell data was performed using Seurat (Wilcox test, p adjusted <0.01), see page 33-34. DESeq2 was only used in the single cell data on a cluster level to check for differential TE subfamilies, see p. 26.

For a fuller understanding of the paper please provide tables of all differentially expressed genes between control and KO mice and GO term analysis of these genes.

We thank the reviewer for this suggestion and have included tables of all differentially expressed genes between control and KO mice (both for bulk and single cell RNAseq: Supplementary Table 1 and 2, respectively).

We have also explored the possibility to perform a GO-analysis of the genes that are detected as differentially expressed in the single-nuclei RNA-seq experiment. However, the low number of genes that are differentially expressed in each cell type makes such an analysis unreliable and misleading.

7. Is there a reason TrustER is not described in the methods section? Please adjust. Also at first mention on page 19 it's written "TrustTER".

We have decided to remove the name TrustTER from the manuscript to avoid confusion on how standalone this pipeline is (see p. 13). We plan to make a separate release of the software in the near future, including additional features such as the possibility to use the analysis for other species. However, this type of software development lies well outside the scope of the current manuscript. Therefore, the single-cell TE approach will be described in the method section (p. 26-27) but the name TrustTER will be removed. All scripts used in this analysis will still be available on GitHub.

8. The authors claim to have conducted "morphological analysis" of microglia. 2D analysis may not be a reliable readout. 3D reconstruction methods are available and should be used in such a setting. In case that the authors do not have access to such methods they should please clarify in the manuscript that they have used 2D morphological analysis and clearly state the shortcomings due to lack of 3-dimensional information.

We thank the reviewer for the comment. We have clarified that the high content screening was performed in 2D on p. 29.

9. On page 21, Iba1 is referred to as "inflammatory factor Iba1". This is misleading, please change to "the pan myeloid cell marker ionized calcium-binding adapter molecule 1 (Iba1) encoded by the Allograft inflammatory factor 1 (Aif1) gene". Please remove the statement that Iba1 is "specific" for microglia. It is just a pan-myeloid cell marker.

We thank the reviewer for the comments. We have modified this statement according to the suggestion from the reviewer on p. 14.

10. When claiming that microglia show "clear signs of activation" it is advisable to show downregulation of homeostatic microglia markers such as P2ry12 or Tmem119 and upregulation of microglial activation markers, such as MHC II and CD68. To support their claim and to back up the title of this study the authors need to provide these analyses.

We did not have success with IHC using the P2ry12, Tmem119 and CD68 antibodies, however, we have analyzed the expression of the P2ry12 and CD68 expression using WB. There were no significant changes in the P2ry12 expression between the Emx1-Cre/Trim28 Ctl and KO animals. There was however an upregulation of CD68 in the cortex of the KO animals (see below), which has been added to Fig 6.

11. Furthermore, microglia activation is associated with increased microglia densities. Please provide data if microglia counts per mm² change in the areas with ERV accumulation.

We have investigated the microglia densities by counting Iba1+ cells per mm². We did not detect any changes in Iba1+ cell densities in cortex between the control and trim28-KO animals (see below). This data has been added in the text on page 14 and in Fig S5a. This goes in line with the mild phenotypic changes seen in the Iba1+ cell morphology in the Trim28-KO animals as well as the microglia cell proportion seen between the Ctl/KO in the single cell sequencing data.

Iba1+ cells per mm² in Ctx

20 fields per animals and area were used for the Iba1+ cell density investigation. Student's t-test revealed no significant changes in cell densities between KO and Ctl animals, either in the cortex or the striatum.

12. On page 24 the authors state that "This study provides direct *in vivo* evidence that aberrant activation of ERVs during brain development triggers neuroinflammation linked to ERV-derived protein aggregation in the adult brain." and they contrast it with correlation studies. However, at this point the authors only show that the presence of IAP-gag and microglia activation coincide, similar to, for example, Amyloid beta aggregates coinciding with Alzheimer's disease in patients. While it is tempting to make the direct connection, there are a number of alternative hypotheses. For example, microglia are activated due to aberrant neurophysiological activity of neurons expressing ERVs or that ERV expression triggers epileptogenic activity that microglia react to. The authors need to be realistic about the correlative nature of their findings and put it in context with relevant alternative hypotheses that they have not explored.

We agree with the reviewer's comment. We have adjusted this statement to:

"This study demonstrates that aberrant activation of ERVs during brain development *in vivo* triggers an inflammatory response linked to the presence of ERV-derived proteins present in aggregate-like structures in the adult brain." (p. 17)

We have also modified statements along the same line on page 19.

13. Several papers relevant to the topic of the current study have been published in the past 12 months (10.1073/pnas.1901283116, 10.1073/pnas.1822164116 and 10.1172/jci.insight.131093). The authors should please discuss these papers and how they relate to the present study.

We have added these three papers into the discussion on p. 18.

14. On page 25, the authors should also put Trim28 in context with other molecular mechanisms with known roles in developmental ERV repression, such as Dnmt1, Dnmt3 (10.1016/j.ydbio.2009.07.017).

We have added a comment on other molecular mechanisms that control TEs during brain development on page 17.

15. When discussing behavioral changes in Trim28-KO mice please put it in context with existing literature including the three studies mentioned under point 12.

We have added these three papers into the discussion on page 18-19.

16. On page 26, the authors note the lack of an interferon response during ERV activation. Similar observations were made in two landmark studies about ERV activation (10.1016/j.immuni.2012.07.018, 10.1038/nature11599). Please put your study in context with these.

We have added these two papers into the new paragraph on the interferon response in the discussion on p. 18-19.

Minor comments

It would have been helpful to provide a pdf document with numbered lines to facilitate the reviewing process.

N/A

Referee #3:

In this manuscript, the authors inactivate the Trim28/KAP1/Tif1beta co-repressor in mouse neuronal cells. In NPCs, in vitro, the inactivation results in increased transcription at a limited number of endogenous retroviruses (ERVs). A similar inactivation in adult neurons in vivo, interestingly, does not result in the reactivations of these ERVs, and the authors show that Trim28 needs to be inactivated during development to reach activity of the ERVs in the adult brain. This prompts the authors to suggest that Trim28, in association with Histone H3 lysine 9 methylation is required at the stem cell stage to allow for more stable repression of the ERVs later in life. This part of the study is followed up by an in-depth analysis at the single-cell level of ERV expression in the adult brain. Finally, the authors show that inactivation of Trim28 is associated with inflammation, possibly caused by strong expression and aggregation of proteins encoded by ERVs.

Overall, this study is thorough and contains several interesting observations. The main concern is that the interpretation of the data focuses exclusively on increased transcription of ERVs in the absence of Trim28, when in fact only a minute fraction of the ERVs get reactivated. Envisioning other possibilities as suggested below would greatly strengthen the paper. In addition, the last figures of the manuscript are somewhat descriptive, but this could probably be adjusted with minor additional experiments.

We would like to thank the reviewer for the overall positive comments on our manuscript. In the new version of the manuscript we have, in response to the reviewer's comments, added new data and modified the text. We honestly feel that this has resulted in a better manuscript that we now hope is ready for publication.

Figure 1 :

(1) A western blot showing the decreased Trim28 protein expression in the NPC KO cells is required.

We thank the reviewer for highlighting the lack of this control experiment. We have now performed WB-experiments that demonstrate the loss of Trim28 protein upon CRISPR-deletion in NPCs. The new data is inserted in Fig 1c and on p. 5.

(2) Panel 1C reports that 13 out of 1153 copies of MERVK10C are upregulated in the KO cells, while Sup Panel 1A shows that 122 TEs out of a total of approximate 3.5 million TE are upregulated in these cells. Likewise, in panel 1E, the heatmaps clearly show that it is only a very minor proportion of the FL-MMERK10C loci that are affected by the TRIM28 inactivation. First, RT-qPCR validation of some of the few activate loci would make the observation more robust.

We have validated the general upregulation of FL-MMERVK10C by qRT-PCR and the results have been added as Supp Fig 1b. To validate unique elements using qRT-PCR is not possible.

Next, given the very low proportion of activated ERVs, the authors may want to consider that the sequences they see activated have common traits other than being ERVs. For example, they may be sites of H3K9 methylation. This could easily be tested by doing peak calling on their Cut and Run data and then confront these peaks with the transcriptome data (as for FL-MMERK10C loci in 1E).

We present H3K9me3-data for all full length MMERVK10-C elements in figure 1F. This analysis clearly shows that many FL-MMERVK10C elements are covered by H3K9me3 - but still not activated by deletion of Trim28. This observation is in line with many findings in the field where this a clear difference between binding of repressive complexes/histone marks and a functional repression as monitored by activation of transcript. While this is a very interesting phenomenon that is worthwhile to investigate further, we feel that this lies well outside the scope of the current study.

(3) Claiming that H3K9 methylation on FL-MMERK10C loci (as shown but the Cut & Run data) links these ERVs to Trim28 binding is a shortcut. This should be rephrased in the text.

We agree with the reviewer and has removed this statement.

(4) In the Sup. Figure 1, the inability of the PCA analysis in distinguishing WT from KO is not a very solid argument in favor of an absence of effect on gene expression. It is rather suggestive of poor reproducibility. This part should be rephrased.

We thank the reviewer for highlighting this. We have re-phrased the text on page 7 to clarify our conclusion from these two PCA-plots: “PCA analyses of differentially expressed protein coding genes and TEs revealed that the Trim28-KO cells separated from control cells based on TE expression rather than gene expression (Fig S1d-e). Together these results demonstrate that Trim28 robustly represses the transcription of ERVs in NPCs but has a marginal direct effect on protein coding genes.”

We have also added a clarifying indication in SFig 1e-f, see figure below.

(5) Examining the bigwigs shows that some of the usual targets of Trim28 do in fact get upregulated in the NPC Trim28 KO cells. The authors may want to look for example at the region around Zfp991 (which incidentally is also extensively methylated on H3K9 according to the Cut & Run data). As this region gives strong Trim28 ChIP signal in other tissues, the effect of Trim28 inactivation on expression of genes in this region may be direct. Other similar regions may exist and could encode genes affecting the fate of the NPCs. This should be discussed.

We agree that several previously reported Trim28 targets are upregulated in the KO-NPCs. We highlight one such example in Fig 1g (BC048671). However, we see much less of this effect in the in vivo experiments (Fig S3d). This is mentioned and discussed on p. 10. We agree that a detailed investigation into this phenomenon might be interesting. However, we feel that this lies outside the scope of the current study.

Figure 3: Examining DNA methylation in the adult brain of the Emx1-Cre (+/-), Trim28-flox (+/+) mice would allow to confirm the hypothesis (Trim28 in stem cells defines DNA methylation later in development). This could be done genome-wide or just by checking some of activated TE by pyrosequencing.

We agree with the reviewer that analysing the status of DNA methylation of TEs in the adult brain of wt and Trim28-mutant mice could be extremely interesting. However, this experiment would have to be performed at single cell-type resolution to be conclusive. Thus, this would require us to either isolate excitatory neurons with FACS prior to DNA-methylation analysis or establish a single-cell DNA-methylation approach that allow for the analysis of TEs. We think this lies well outside the scope of the current study – but this line of research will be something that we will pursue in the future.

In addition, the authors claim that Sup. Figure 3 demonstrates that increased ERV transcription affects gene expression. This seems an overstatement, as the figure, strictly speaking, only shows that inactivation of Trim28 in NPCs affects gene expression later in life. This must be rephrased.

We agree with the reviewer that this was an overstatement. We have adjusted the text accordingly: “This demonstrates that the loss of Trim28 during brain development causes substantial downstream effects on gene expression”

Figure 6: The presence of aggregates in adult tissues from adult Emx1-Cre (+/-), Trim28-flox (+/+) is an interesting observation. First, the authors need to verify that neither their primary nor their secondary antibodies stick to aggregates.

We thank the reviewer for the comments. We use both the IAP antibody and our secondary antibodies routinely in our lab and have not observed any of these antibodies to stick to aggregates. However, to avoid any overstatement of the IAP-IHC we have rephrased the text. In some instances we have removed the word completely and in other parts of the text we refer to it as “aggregation-like structures”.

Next, to gain mechanistic insight on how aggregates cause inflammation, the authors may want to check for activation of ER stress. The presence of ER stress is suggested by the increased expression of the ATF6 targets NR4A3 and EGR1 in the Emx RNA-seq data. To demonstrate this more directly, the authors could use their extracts from panel 6E with an anti-phosphoATF6 antibody to monitor maturation/cleavage of this protein.

As recommended by the reviewer, we have looked into the data with ER-stress in mind. The expression of the ATF6 targets NR4A3 and EGR1 are, like the reviewer points out, slightly increased in 2 out of 3 KO animals, but are not significantly different to the ctl. We did however perform a western blot analysis for pATF6 as suggested and detected the full-length protein in both ctl and KO animals where the protein was significantly higher expressed in the KO animals (n=5 for both groups). The cleaved protein was not detected. Although the reviewer is raising an interesting point, to properly investigate if the presence of elevated ERV expression is causing inflammation via ER-stress would make a whole independent study and, in our opinion, lies outside the scope of this study.

Minor points:

(1) Manual examination of the provided bigwigs showed increased transcriptional activity at several SINEs, which is in contrast with the description in the text. This could be due to poor management of the multi-mappers. Nevertheless, this should be checked.

We thank the reviewer for this comment. We checked the datasets for which we have reported no significant SINE upregulation, and found indeed that there seem to be lot of upregulated SINE elements. However, most of them do not have a consistent trend between conditions, given that p-values were not able to calculate (or were very high) in most of them. Given the inconsistencies in our data and the well-established experimental challenge there is to analyze the expression of SINE-elements using standard RNAseq data, we have decided to not pursue this further. We have, however, modified the statement on p. 6 that relates to the expression of other TEs

(2) RepeatMasker contains about 5 million TEs, Panel 1A examines only about 3.5 million TE. How were they selected?

We thank the reviewer for noticing the lack of describing the TE selection strategy for panel 1A. It has been added to the methods section on p. 23-24. Briefly, for the analysis of TE subfamilies, we used a custom-made GTF file from the Tetranscripts' authors (with ~3.7 million entries). We used the same GTF files for the read quantification per element. The annotation tables were parsed to filter out low complexity and simple repeats, rRNA, scRNA, snRNA, srpRNA and tRNA.

Dear Johan,

Thanks for submitting your revised manuscript to The EMBO Journal. Your study has now been seen by the three referees and their comments are provided below. As you can see from the comments, the referees appreciate the introduced changes and support publication here. Referee #2 has a few suggestions that I would like to ask you to take into consideration in a final revision. I think the referee has a good point regarding the use of the term inflammation. Let me know if we need to discuss this issue further. When you submit your revised manuscript will you also please take note of the following points:

- Please add keywords
- Re-label Declaration of Interests as COI
- Figure callouts: There is a callout to Fig. S3A, but there is no such figure. Fig EV5D callout is missing. Fig EV4 is called out as Fig S4.
- Methods needs correcting to Materials and Methods.
- Figures 6 A & F please add scale bars.
- Regarding the appendix file, the tables have no names and should also be referenced in the manuscript file. To me this is data that could be part of M&Ms and I wonder if we could simply add callouts from the M&Ms to the tables.
- Can you also add a reference in the text to the Metadata worksheet- this worksheet should also be re-named as a table.
- Please also split the source data into individual files (one file per figure)
- I have asked our publisher to do their pre-publication checks on the paper. They will send me the file within the next few days. Please wait to upload the revised version until you have received their comments.
- We include a synopsis of the paper (see <http://emboj.embopress.org/>). Please provide me with a general summary statement and 3-5 bullet points that capture the key findings of the paper.
- We also need a summary figure for the synopsis. The size should be 550 wide by [200-400] high (pixels). You can also use something from the figures if that is easier.

That should be all!

With best wishes

Karin

Karin Dumstrei, PhD

Senior Editor
The EMBO Journal

Further information is available in our Guide For Authors:

The revision must be submitted online within 90 days; please click on the link below to submit the revision online before 8th Apr 2021.

Referee #1:

I was already quite enthusiastic about this manuscript in the previous round of review. I raised several critiques of the text, and i feel that the authors have been responsive to my comments. I think this is good to go.

Referee #2:

Reviewer comments Jönsson et al. EMBO

The authors have implemented many of the suggested reviewer comments and made some valuable additions to an overall good manuscript. However, the reviewer still has misgivings with the term inflammation used throughout the manuscript. On the pages 14 and 15 that inform the inclusion of the term "inflammation" into the title of the manuscript the authors start out their data description with:

"aberrant expression of ERVs and other TEs have been linked to inflammation (Hurst and Magiorkinis, 2015; Ishak et al., 2018; Lim et al., 2015; Roulois et al., 2015; Saleh et al., 2019; Tam et al., 2019a; Thomas et al., 2017)."

And conclude on page 15 with:

"The inflammation was therefore spatially restricted to the area with increased ERV expression." The data that they show is, as the authors rightfully state, about microglia activation. The term inflammation requires the detection of cytokines at protein levels. This was not performed. The RNA-Seq analysis in extended Figure 5 shows the expression of some viral defense genes, but none of them reaches statistical significance. Again, all the reviewer sees is microglia activation shown through the limited morphological data and protein data of the activation marker CD68. In the interest of clarity the reviewer has to insist for the usage of terms that reflect the data. Of course there is some literature on inflammation in ERV overexpression, but if, as per the title, the main point of the paper is to show inflammation, but all that the authors show is microglia activation then we should be explicit about the emperor's new clothes. Please don't take this as unjustified criticism. The reviewer does appreciate the manuscript and there is no doubt in its importance, but the data interpretation must not be exaggerated. Please adjust your terminology and also fix the presentation of the western blot data as suggested below as well as the other comments.

Page 15 - when introducing CD68 please put it in biological context as a lysosomal protein upregulated in activated microglia.

Figure 6d - Also, the CD68 Western blot is not well chosen, since the control and the KO sample seems to be cropped out. Thus, it is not obvious for the reader if the difference in band intensity is due to different exposure times or if it's real. Please provide a western blot where you show several control and KO bands side by side. The same goes for the IAP Gag western blot.

Figure 6e - As for the IAP gag western blot. Do the authors observe differences in the band intensities of the bands for the cleaved IAP gag products (see Figure 3b in <https://doi.org/10.1038/ng1353>)? Please clarify.

Referee #3:

The authors have properly addressed most of the essential issues, with several high-quality additions.

Response to comments from the reviewers

Referee #1:

I was already quite enthusiastic about this manuscript in the previous round of review. I raised several critiques of the text, and i feel that the authors have been responsive to my comments. I think this is good to go.

- We thank the reviewer for all the input.

Referee #2:

Reviewer comments Jönsson et al. EMBO

The authors have implemented many of the suggested reviewer comments and made some valuable additions to an overall good manuscript. However, the reviewer still has misgivings with the term inflammation used throughout the manuscript. On the pages 14 and 15 that inform the inclusion of the term "inflammation" into the title of the manuscript the authors start out their data description with:

"aberrant expression of ERVs and other TEs have been linked to inflammation (Hurst and Magiorkinis, 2015;Ishak et al., 2018; Lim et al., 2015; Roulois et al., 2015; Saleh et al., 2019; Tam et al., 2019a;Thomas et al., 2017)."

And conclude on page 15 with:

"The inflammation was therefore spatially restricted to the area with increased ERV expression."

The data that they show is, as the authors rightfully state, about microglia activation. The term inflammation requires the detection of cytokines at protein levels. This was not performed. The RNA-Seq analysis in extended Figure 5 shows the expression of some viral defense genes, but none of them reaches statistical significance. Again, all the reviewer sees is microglia activation shown through the limited morphological data and protein data of the activation marker CD68. In the interest of clarity the reviewer has to insist for the usage of terms that reflect the data. Of course there is some literature on inflammation in ERV overexpression, but if, as per the title, the main point of the paper is to show inflammation, but all that the authors show is microglia activation then we should be explicit about the emperor's new clothes. Please don't take this as unjustified criticism. The reviewer does appreciate the manuscript and there is no doubt in its importance, but the data interpretation must not be exaggerated. Please adjust your terminology and also fix the presentation of the western blot data as suggested below as well as the other comments.

- We have now removed the word inflammation throughout the manuscript.

Page 15 - when introducing CD68 please put it in biological context as a lysosomal protein upregulated in activated microglia.

- **We have added this information on page 15.**

Figure 6d - Also, the CD68 Western blot is not well chosen, since the control and the KO sample seems to be cropped out. Thus, it is not obvious for the reader if the difference in band intensity is due to different exposure times or if it's real. Please provide a western blot where you show several control and KO bands side by side. The same goes for the IAP Gag western blot.

- **We can ensure the reviewer that all samples were photographed at the same time with the same exposure. All uncropped photos for our western blots, together with a description of loading details, are to be found in the source data files.**

Figure 6e - As for the IAP gag western blot. Do the authors observe differences in the band intensities of the bands for the cleaved IAP gag products (see Figure 3b in <https://doi.org/10.1038/ng1353>)? Please clarify.

- **We did not analyze the cleaved IAP gag products.**

Referee #3:

The authors have properly addressed most of the essential issues, with several high-quality additions.

- **We thank the reviewer for all the input.**

Dear Johan,

Thanks for submitting your revised manuscript to The EMBO Journal. I have now had a chance to take a careful look at everything and I appreciate the introduced changes.

I am therefore very pleased to accept the manuscript for publication here. Congratulations on a nice study

with best wishes

Karin

Karin Dumstrei, PhD
Senior Editor
The EMBO Journal

Please note that it is EMBO Journal policy for the transcript of the editorial process (containing referee reports and your response letter) to be published as an online supplement to each paper. If you do NOT want this, you will need to inform the Editorial Office via email immediately. More information is available here: https://emboj.embopress.org/about#Transparent_Process

Your manuscript will be processed for publication in the journal by EMBO Press. Manuscripts in the PDF and electronic editions of The EMBO Journal will be copy edited, and you will be provided with page proofs prior to publication. Please note that supplementary information is not included in the proofs.

Should you be planning a Press Release on your article, please get in contact with embojournal@wiley.com as early as possible, in order to coordinate publication and release dates.

If you have any questions, please do not hesitate to call or email the Editorial Office. Thank you for your contribution to The EMBO Journal.

Corresponding Author Name: Johan Jakobsson

Journal Submitted to: The EMBO journal

Manuscript Number: EMBOJ-2020-106423